# CoPINN: Cognitive Physics-Informed Neural Networks

**Siyuan Duan** [* 1]  **Wenyuan Wu** [* 1]  **Peng Hu** [1]  **Zhenwen Ren** [2]  **Dezhong Peng** [1 3]  **Yuan Sun** [1 3]

## Abstract

Physics-informed neural networks (PINN) aim to constrain the outputs and gradients of deep learning models to satisfy specified governing physics equations, which have demonstrated significant potential for solving partial differential equations (PDEs). Although existing PINN methods have achieved pleasing performance, they always treat both easy and hard sample points indiscriminately, especially ones in the physical boundaries. This easily causes the PINN model to fall into undesirable local minima and unstable learning, thereby resulting in an Unbalanced Prediction Problem (UPP). To deal with this daunting problem, we propose a novel framework named C̲ognitive P̲hysics-I̲nformed N̲eural N̲etworks (**CoPINN**) that imitates the human cognitive learning manner from easy to hard. Specifically, we first employ separable subnetworks to encode independent one-dimensional coordinates and apply an aggregation scheme to generate multi-dimensional predicted physical variables. Then, during the training phase, we dynamically evaluate the difficulty of each sample according to the gradient of the PDE residuals. Finally, we propose a cognitive training scheduler to progressively optimize the entire sampling regions from easy to hard, thereby embracing robustness and generalization against predicting physical boundary regions. Extensive experiments demonstrate that our CoPINN achieves state-of-the-art performance, particularly significantly reducing prediction errors in stubborn regions. The code is available at this repository: https://github.com/siyuancncd/CoPINN.

---

[*]Equal contribution  [1]College of Computer Science, Sichuan University, Chengdu, China. [2]Southwest University of Science and Technology, Mianyang, China. [3]National Key Laboratory of Fundamental Algorithms and Models for Engineering Numerical Simulation, Sichuan University, Chengdu, China. Correspondence to: Yuan Sun <sunyuan_work@163.com>.

*Proceedings of the 42ˢᵗ International Conference on Machine Learning*, Vancouver, Canada. PMLR 267, 2025. Copyright 2025 by the author(s).

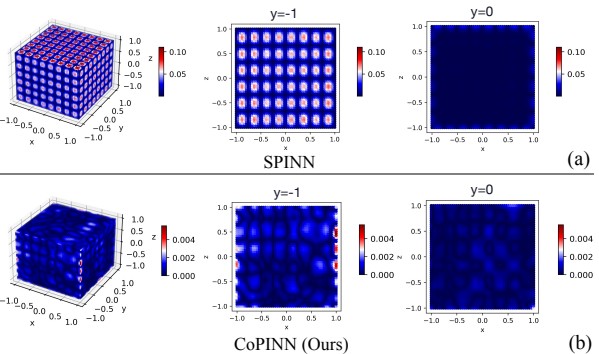

*Figure 1.* The 2D and 3D visualization of the absolute error between the predicted and exact values. The left graph illustrates the absolute error of the entire three-dimensional space. The middle graph demonstrates the absolute error of the boundary when $y = -1$, while the right graph displays the absolute error of the cross-section when $y = 0$. (a) The absolute error of SPINN (SOTA) (Cho et al., 2023) on the Helmholtz equation. These results indicate that SPINN exhibits significantly larger errors near the physical boundary region compared to the middle region, which reveals the Unbalanced Prediction Problem (UPP). (b) The absolute error of our CoPINN on the Helmholtz equation, which shows that CoPINN maintains consistent small absolute errors both near the physical boundary and in the middle region.

## 1. Introduction

As a type of equation in mathematics that describes the relationship between functions of multiple variables (Evans, 2022), partial differential equations (PDEs) usually involve partial derivatives of unknown functions and describe the laws of change of natural phenomena in disciplines such as physics, chemistry, and economics. In recent years, solving PDEs has a wide range of applications in many fields of science and engineering (Roubíček, 2013; Li et al., 2025), which are crucial for understanding and uncovering the underlying physical laws. Therefore, a large number of traditional numerical methods have been developed to solve PDEs, such as finite element methods (Jagota et al., 2013), finite difference methods (Thomas, 2013), and finite volume methods (Barth et al., 2018). Unfortunately, these methods are resource-intensive and mesh-dependent, thus often requiring a huge time cost to solve such complex systems, for example, the Navier-Stokes equations (Zou et al., 2024a). Thanks to the powerful nonlinear representation capabilities of deep learning, they are also widely used to solve partial

differential equations. As a mainstream solution method, physics-informed neural networks (PINN) (Raissi et al., 2019; Long et al., 2018; Kim et al., 2021; Wright et al., 2022) have become a promising and effective alternative to numerical methods, which aim to embed physical prior knowledge into neural networks to improve flow field prediction capability.

Compared with traditional data-driven methods, PINN can not only enhance the generalization ability of the model but also improve the prediction accuracy of the model in data-scarce environments. Thus, a large number of PINN methods have been proposed, which mainly focus on the four techniques: i.e., neural network architectures, optimization schemes, loss re-weighting, and adaptive sampling. The first category focuses on designing the architecture of the neural network part of PINN, such as uncertainty estimation (Yang et al., 2021; Wu et al., 2025) and solving multi-dimensional PDEs (Cho et al., 2023). The second category, from the optimization perspective, designs the loss function for PINN, such as predicting evolutionary equations (Li et al., 2024) and region optimization (Wu et al., 2024). The third category aims to construct the loss re-weighting strategy to achieve optimization of specific loss terms (Wang et al., 2022) or training points (Xiang et al., 2022). The last category dynamically adjusts the distribution of training data and gives priority to dense sampling in areas with large residuals or drastic solution changes (such as boundary layers, shock waves, etc.), thereby improving the accuracy of the model in key areas and the overall training efficiency (Wu et al., 2023; Daw et al., 2023).

Although these methods have achieved promising performance, they still face some challenges in solving PDEs. To be specific, almost all of them implicitly assume that all training data are equally important while ignoring the differences between samples. In other words, the existing PINN methods always treat both easy and hard samples equally while ignoring the effect of physical boundaries on the learning difficulty of the data. This easily results in unstable learning and falls into poor local minima, thereby leading to an Unbalanced Prediction Problem (UPP). As shown in Figure 1, for the state-of-the-art PINN method (i.e., SPINN (Cho et al., 2023) ), we visualize the absolute error between the predicted value and the exact value on the Helmholtz equation. According to their prediction performance, we can observe that the error values in stubborn regions (such as physical boundaries) are much larger than those in the middle smooth regions. This shows that the biggest challenges in solving PDEs are mainly in hard physical boundary regions, rather than easy smooth regions.

Inspired by the human cognitive learning process, self-paced learning (SPL) (Jiang et al., 2015) was proposed to train these samples in the training set from easy to hard. Thanks

to such a gradual learning paradigm, SPL can enhance the generalization ability according to the difficulty differences of these samples. However, although SPL has been explored in numerous areas such as clustering (Zhou et al., 2023; Bai et al., 2024), classification (Yuan et al., 2024; Chen et al., 2024), retrieval (Sun et al., 2024a;b; Pu et al., 2025), and domain generalization (Zhao et al., 2024), it has never been touched in the field of PINN. The greatest challenge is how to evaluate the difficulty of samples from the physical boundary and ones from the smooth region.

To overcome the above challenge, we propose a novel Cognitive Physical Informed Neural Network (**CoPINN**) that overcomes the problem of difficult optimization of samples in physical boundary regions. As shown in Figure 2, our CoPINN effectively emulates human cognitive learning, beginning with easier regions and progressively advancing to more challenging ones, thereby endowing the model with generalization in difficult regions. To be specific, we first adopt the separable sub-networks to encode independent one-dimensional coordinates instead of using a single MLP for all multi-dimensional coordinates, thereby reducing the computational complexity of solving PDEs. Then, we utilize an aggregation scheme to obtain the multi-dimensional predicted physical variables. Afterward, in the training process, CoPINN dynamically evaluates the difficulty of predicting each sample by the gradient magnitude of the PDE residuals. Finally, we present a cognitive training scheduler to adaptively optimize the PINN model from easy to hard, thereby endowing it the robustness and generalization against predicting physical boundary regions. In summary, the main contributions of this paper are as follows:

- We reveal and study an untouched yet pervasive significant problem in PINN, termed the unbalanced prediction problem (UPP). Unlike previous PINN methods that always treat both easy and hard samples equally, we propose a novel cognitive PINN framework to alleviate the negative effect of the hard samples in stubborn regions during the learning process.

- We propose a PINN model with SPL that measures the sample-level difficulty and promotes the neural network to fit samples from easy to hard in solving PDEs. To the best of our knowledge, our CoPINN could be the first work that leverages SPL to enhance the PINN performance in difficult regions.

- We conduct a series of experiments on multiple widely used PDE equations. These numerical results demonstrate that the proposed CoPINN can consistently outperform seven state-of-the-art PINN methods by a considerable margin under different numbers of collocation points.

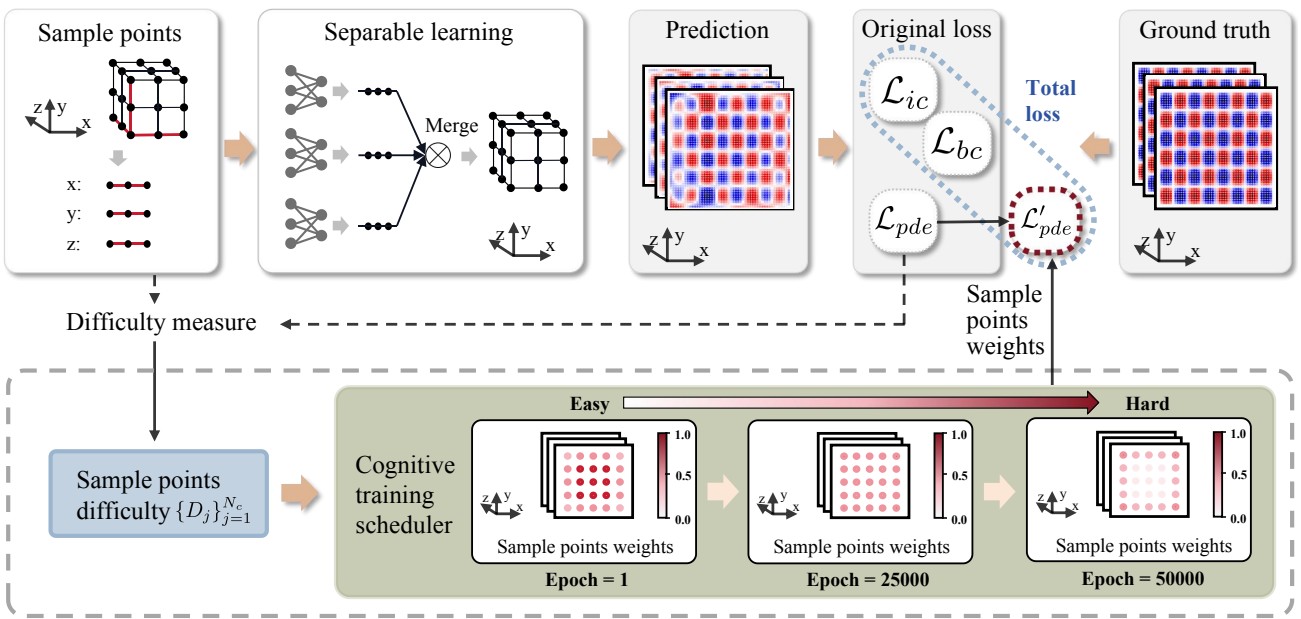

*Figure 2.* The frameworks of CoPINN. The separable sub-networks are used to encode independent one-dimensional coordinates. Then, an aggregation scheme is applied to obtain the multi-dimensional predicted physical variables. During the training process, CoPINN dynamically evaluates the difficulty of predicting each sample based on the gradient magnitude of the PDE residuals. Finally, a cognitive training scheduler is employed to adaptively optimize the model from easy to hard.

## 2. Method

### 2.1. Notations and Motivation

Without loss of generality, we denote the general partial differential equation (PDE) as the following formula, i,e.,

$$\begin{cases} u_t = \mathcal{N}(u), \ x \in \Omega, t \in \Gamma, \\ u(x,0) = u_0(x), \ x \in \Omega, \\ u(x,t) = g(x,t), \ x \in \partial\Omega, \ t \in \Gamma, \end{cases} \quad (1)$$

where $u(x,t)$ denotes the hidden solution, $t$ and $x$ represent temporal and spatial coordinates respectively, $\Omega$ represents an open, bounded domain with smooth boundary $\partial\Omega$, $\Gamma$ represents a time domain, $\mathcal{N}$ denotes a differential operator, and $g(x,t)$ is a known boundary condition function. Our goal is to develop a PINN method to obtain approximate solutions with reasonable accuracy for Equation (1) through neural network fitting. Although existing PINN methods achieve promising solving performance, all of them overlook the effect of physical boundaries on the learning difficulty of data, which could easily lead to unstable learning and get stuck in a poor local minimum. Therefore, this inevitably leads to the Unbalanced Prediction Problem (UPP) for solving PDEs, which is shown in Figure 1. To address this, we propose a novel Cognitive Physics-Informed Neural Networks (CoPINN). In this paper, we first introduce separable learning in Section 2.2, followed by a description of our learning objective in Section 2.3, and conclude with the cognitive learning scheduler in Section 2.4 and Section 2.5.

### 2.2. Separable Learning

Traditional PINN usually utilizes an MLP architecture based on spatiotemporal coordinates to represent the solution function, where the input corresponds to spatiotemporal coordinates and the output represents the associated solution quantity. For each training point, computing the PDE residual loss necessitates multiple forward and backward propagations. As the number of training points (collocation points) increases, particularly for higher-dimensional or more complex solutions, the computational burden escalates. To address this, we follow SPINN (Cho et al., 2023) and adopt a separable architecture to handle high-dimensional PDEs efficiently. Specifically, we construct $d$ independent sub-networks (MLPs), each of which takes a single one-dimensional coordinate as input. Each sub-network $f^{(\theta_i)}$ is a vector-valued function that maps the $i$-th coordinate to an $r$-dimensional feature representation. Afterward, we aggregate these feature representations to obtain the final prediction as follows,

$$\hat{u}(x_1, x_2, ..., x_d) = \sum_{j=1}^{r} \prod_{i=1}^{d} f_j^{(\theta_i)}(x_i), \quad (2)$$

where $\hat{u}$ is the predicted solution function, $x_i \in \mathbb{R}$ is a coordinate of $i$-th axis, and $f_j^{(\theta_i)}$ represents the $j$-th element of $f^{(\theta_i)}$. In practice, the input coordinates are provided in batches during both training and inference. To this end, we extend Equation (2) to allow the neural network to process

mini-batches for efficient training. Specifically, let $N$ input coordinates, i.e., training points, be sampled from each axis. The sampling resolutions for different axes need not be identical. The input coordinates $\mathbf{X} \in \mathbb{R}^{N \times d}$ form a matrix. Consequently, we could obtain the batchified feature representation $F \in \mathbb{R}^{N \times r \times d}$. Further, Equation (2) could be extended as the following formula, i.e.,

$$\hat{U}(X_{:,1}, X_{:,2}, ..., X_{:,d}) = \sum_{j=1}^{r} \bigotimes_{i=1}^{d} F_{:,j,i}, \qquad (3)$$

where $\hat{U} \in \mathbb{R}^{N \times N \times, ..., \times N}$ is the discretized solution vector, $\bigotimes$ represents outer product, $F_{:,:,i} \in \mathbb{N}^{N \times r}$ denotes the $i$-th frontal slice matrix of vector $F$, and $F_{:,j,i} \in \mathbb{R}^N$ is the $j$-th column of the matrix $F_{:,:,i}$. Note that, because each coordinate axis is encoded separately by this scheme, compared with the traditional multi-dimensional coordinate encoding, the memory and time consumption of calculation are greatly reduced.

## 2.3. Problem Formulation

In classical PINN (Raissi et al., 2019), a deep neural network is trained to approximate the solution of a PDE. The idea behind PINN is to embed the underlying physics information (i.e., PDEs) directly into the loss function, alongside traditional data-driven loss terms like initial/boundary conditions and data points. Mathematically, the objective function $\mathcal{L}(\hat{u}^\theta)$ of the vanilla PINN could be represented as

$$\mathcal{L}(\hat{u}^{(\theta)}) = \lambda_{pde}\mathcal{L}_{pde} + \lambda_{ic}\mathcal{L}_{ic} + \lambda_{bc}\mathcal{L}_{bc},$$
$$\mathcal{L}_{pde} = \int_\Gamma \int_\Omega ||\mathcal{N}\left[\hat{u}^{(\theta)}\right](x,t)||^2 \, dx \, dt,$$
$$\mathcal{L}_{ic} = \int_\Omega ||\hat{u}^{(\theta)}(x,0) - u_{ic}(x)||^2 \, dx, \qquad (4)$$
$$\mathcal{L}_{bc} = \int_\Gamma \int_\Omega ||\mathcal{B}\left[\hat{u}^{(\theta)}\right](x,t) - u_{bc}(x,t)||^2 \, dx \, dt,$$

where $\hat{u}^{(\theta)}$ denotes the predicted solution function, $x$ represents space coordinates, $t$ represents time coordinates $\Omega$ represents an input domain, $\Gamma$ represents an time domain, $\mathcal{N}$ and $\mathcal{B}$ denote generic differential operators, and $u_{bc}$ and $u_{ic}$ are initial, boundary conditions, respectively. $\lambda_*$ are balancing factors for each loss term.

In Equation (4), the loss function in PINN often includes terms that ensure the neural network satisfies the initial conditions and boundary conditions of the physical problem being modeled. The network's output at every point must satisfy the physics (through the PDE residual) and conform to the known physical constraints (through initial and boundary conditions). Since there are some mutations in the samples of the physical boundaries, this undoubtedly increases the difficulty of learning these stubborn points.

Previous PINN methods often implicitly assume that all training data is equally important while ignoring their differences in learning difficulty. It is unreasonable, which has been proven in Figure 1. To tackle this UPP, we present a CoPINN method to predict the solution of PDE, which imitates human cognitive learning to gradually incorporate more challenging samples from easy to hard during the process of training the PDE loss. Therefore, the learning objective of CoPINN can be formulated as follows:

$$\min_\theta \mathcal{L}(\hat{u}^{(\theta)}) = \min_\theta \left( \lambda_{pde} \int_\Gamma \int_\Omega v_i \cdot ||\mathcal{N}\left[\hat{u}^{(\theta)}\right](x,t)||^2 dx dt \right.$$
$$+ \lambda_{ic} \int_\Omega ||\hat{u}^{(\theta)}(x,0) - u_{ic}(x)||^2 dx$$
$$\left. + \lambda_{bc} \int_\Gamma \int_\Omega ||\mathcal{B}\left[\hat{u}^{(\theta)}\right](x,t) - u_{bc}(x,t)||^2 dx dt \right),$$

$$(5)$$

where $v_i \in [0,1]$ represents the weight assigned to sample point $\mathbf{x}_i$, which dynamically changes as the training proceeds.

## 2.4. Difficulty Evaluation

In traditional SPL (Wang et al., 2021b; Soviany et al., 2022; Shrivastava et al., 2016), difficulty-level is often based on predefined heuristics, like the training loss or certain statistical properties of the data, which might not fully capture the complexities of the solution space in the context of PINN. Utilizing just the training loss (disparity between predictions and ground truth) could not provide a nuanced enough measure for difficult regions of the solution space, especially in PDEs that exhibit phenomena like shock waves, singularities, or boundary layers, which exhibit rapid solution changes. These regions are physically important, thus, we should handle them with greater care during training. To this end, we regard the gradient of the PDE residual as a measure of sample difficulty. By measuring the spatial or temporal gradient of the residual, we can identify where these phenomena occur. Areas with high gradients are indicative of regions where the solution changes rapidly, meaning the network might need additional effort to approximate the solution. In addition, flatter regions (low gradient) suggest simpler behaviors where the network could focus less on achieving high accuracy. To be specific, we dynamically adjust the difficulty measure based on the current state of the model, thereby enabling SPL to be much more flexible and responsive. Instead of relying on a static, predefined difficulty metric, the model can continuously adapt the focus during training, emphasizing the more challenging parts of the solution. To be specific, the difficulty of $k$-th sample in $i$-th epoch could be expressed as

$$D_k^i = \left\| \frac{\partial \mathcal{L}_{pde}^i}{\partial \mathbf{x}_k^i} \right\|_2. \qquad (6)$$

## 2.5. Cognitive Training Scheduler

As shown in Figure 1 (a), samples from regions with smooth variations in physical quantities are typically easier to handle by current PINN methods. Overemphasizing these samples while neglecting those from more challenging regions can impede the model's ability to accurately fit regions with abrupt changes in physical quantities. Effectively learning from regions with significant variations in physical variables is crucial for accurately solving partial differential equations (PDEs). To address this challenge, after identifying where physical phenomena occur, such as shocks or singularities, through a difficulty evaluation, we propose a cognitive training scheduler to gradually pay more attention to those challenging regions. Early in training, when the network has less capability to fit difficult regions, giving it more weight on easy regions helps the model warm up by learning the basic structure of the solution. As training continues, the network becomes more capable, and the scheduler directs the focus to harder regions. This is essential to prevent the model from overfitting to the simple regions and underfitting the complex, critical regions. This dynamic scheduling enables the model to learn progressively, ensuring that it's exposed to both simple and complex regions of the solution, which is vital for accurate and robust generalization.

To this end, we dynamically adjust the weights assigned to easy and difficult samples. As illustrated in Figure 2, during the initial epoch, easier samples (e.g., those in the middle region) are assigned higher weights compared to more difficult samples (e.g., those in boundary regions). Throughout the training process, the weight difference between easy and difficult samples gradually diminishes. By the midpoint of training, the weights for both easy and difficult samples equalize. Subsequently, the weights for difficult samples incrementally exceed those for easy samples. To be specific, during the training process, from the first epoch to the final one, the weight assigned to the easiest samples decreases from one to zero. For a training phase with $N_e$ epochs, we expect the weight of the easiest sample to be $1$ in the first epoch and $0$ in $N_e$ epochs. Then, for the easiest sample, the magnitude of the weight modifications in each epoch can be calculated as follows:

$$\tau_e = \frac{v_e^1 - v_e^{N_e}}{N_e} = \frac{1}{N_e}, \tag{7}$$

where $v_e^1$ and $v_e^{ne}$ are the weights assigned to the easiest sample in the first and final epoch, respectively, with $v_e^1 = 1$ and $v_e^{N_e} = 0$. Therefore, the weights assigned to the easiest and hardest samples in epoch $i$, denoted as $v_e^i$ and $v_h^i$, are computed as follows:

$$v_e^i = v_e^1 - \tau_e \cdot (i-1) = 1 - \frac{i-1}{N_e}, \tag{8}$$

$$v_h^i = v_h^1 + \tau_e \cdot (i-1) = 0 + \frac{i-1}{N_e}. \tag{9}$$

Once $v_e^i$ and $v_h^i$ are calculated, the weight of the $j$-th most easy sample in the $i$-th epoch can be calculated as follows:

$$\delta_j^i = (v_e^i - v_h^i) \cdot \frac{D_j^i - D_e^i}{D_h^i - D_e^i}, \tag{10}$$

where $D_j^i$ denotes the difficulty measure of $j$-th most easy sample, $D_e^i$ represents the difficulty measure of the easiest sample, and $D_h^i$ represents the difficulty measure of the hardest sample, in $i$-th epoch. The difficulty value is in the range $[0, +\infty]$. Afterward, we can calculate the weight $v_j^i$, for each sample in the $i$-th epoch, where $j$ signifies the sample ranked $j$-th in terms of difficulty measure. It could be represented as

$$v_j^i = v_e^1 - \tau_e \cdot (i-1) - \beta \cdot \delta_j^i, \tag{11}$$

where $\beta$ is a hyperparameter to adjust the weight variety for each sample in each epoch.

Note that $\delta_j^i$ can be computed either globally or locally. When computed globally, $\delta_j^i$ is computed over the entire dataset at once for each epoch. This approach assigns unique weights to samples based on their difficulty rank across the entire dataset, thereby providing precise weight calculation. However, computing $\delta_j^i$ globally can be computationally expensive as it necessitates storing the difficulty measure for each sample until the end of an epoch, limiting its scalability for large datasets. To address this limitation, we suggest computing $\delta_j^i$ locally, which means computing it in one batch size.

## 2.6. Weight Analysis

To further analyze the proposed cognitive training scheduler, we plot the weight trends for different values of $\beta$ in Figure 3. The results clearly show that the weight of the easiest samples decreases as training progresses and the number of iterations increases. Moreover, from Figure 3 (a), (b), to (c), it can be observed that as $\beta$ decreases, the weight of the more difficult samples gradually increases. If there is only a single loss term, the approach illustrated in Figure 3 (a) is often more efficient. This is because it initially focuses on learning easy samples, progressively incorporating difficult samples as the number of iterations increases, thereby eventually covering all samples. However, our method includes three loss terms: $\mathcal{L}_{pde}$, $\mathcal{L}_{ic}$, and $\mathcal{L}_{bc}$. Adopting the self-paced learning strategy shown in Figure 3 (a) may cause the neural network to forget the easy samples learned in the early stages and shift its focus entirely to the difficult samples. To mitigate this issue, we recommend setting $\beta$ to less than $0.5$. A detailed parameter analysis of this choice is provided in Section 3.3.

*Table 1.* Full results of the Helmholtz, (2+1)-d Klein-Gordon and Diffusion equation. $N_c$ is the number of collocation points. Due to the increase in training points, some methods are out-of-memory (O/M). The best and the second-best results are highlighted in **boldface** and underlined, respectively.

| Equation | Evaluation Metric | | $RL_2$ | | | | | $RMSE$ | | | | |
|---|---|---|---|---|---|---|---|---|---|---|---|---|
| | Methods | $N_c$ Ref. | $16^3$ | $32^3$ | $64^3$ | $128^3$ | $256^3$ | $16^3$ | $32^3$ | $64^3$ | $128^3$ | $256^3$ |
| Helmholtz | PINN | JCP'2019 | 0.9819 | 0.9757 | 0.9723 | O/M | O/M | 0.3420 | 0.3398 | 0.3386 | O/M | O/M |
| | gPINN | CMAME'2022 | 0.3852 | 0.3255 | 0.4008 | O/M | O/M | 0.1342 | 0.1133 | 0.1396 | O/M | O/M |
| | AHD-PINN | IJCAI'2024 | 0.2108 | 0.1903 | 0.1871 | O/M | O/M | 0.0734 | 0.0663 | 0.0652 | O/M | O/M |
| | SPINN | NeurIPS'2023 | 0.1177 | 0.0809 | 0.0592 | 0.0449 | 0.0435 | 0.0410 | 0.0282 | 0.0206 | 0.0156 | 0.0151 |
| | SPINN (m) | NeurIPS'2023 | 0.1161 | 0.0595 | 0.0360 | 0.0300 | 0.0311 | 0.0404 | 0.0207 | 0.0125 | 0.0104 | 0.0108 |
| | RoPINN | NeurIPS'2024 | 0.4059 | 0.3338 | O/M | O/M | O/M | 0.1414 | 0.1162 | O/M | O/M | O/M |
| | FPINN | NN'2025 | 0.3862 | 0.3502 | 0.3097 | O/M | O/M | 0.1345 | 0.1220 | 0.1079 | O/M | O/M |
| | CoPINN | Ours | **0.0172** | **0.0050** | **0.0016** | **0.0007** | **0.0006** | **0.0040** | **0.0015** | **0.0004** | **0.0002** | **0.0002** |
| | IMP. | - | 85% | 92% | 96% | 98% | 98% | 90% | 93% | 97% | 98% | 98% |
| (2+1)-d Klein-Gordon | PINN | JCP'2019 | 0.0343 | 0.0281 | 0.0299 | O/M | O/M | 0.0218 | 0.0178 | 0.0190 | O/M | O/M |
| | gPINN | CMAME'2022 | 0.0108 | 0.0025 | O/M | O/M | O/M | 0.0069 | 0.0016 | O/M | O/M | O/M |
| | SPINN | NeurIPS'2023 | 0.0193 | 0.0060 | 0.0045 | 0.0040 | 0.0039 | 0.0123 | 0.0038 | 0.0029 | 0.0025 | 0.0025 |
| | SPINN (m) | NeurIPS'2023 | 0.0062 | 0.0020 | 0.0013 | 0.0008 | 0.0009 | 0.0039 | 0.0013 | 0.0008 | 0.0005 | 0.0006 |
| | AHD-PINN | IJCAI'2024 | 0.0133 | 0.0082 | 0.0147 | O/M | O/M | 0.0084 | 0.0052 | 0.0093 | O/M | O/M |
| | RoPINN | NeurIPS'2024 | 0.1925 | 0.1806 | 0.1740 | O/M | O/M | 0.1223 | 0.1147 | 0.1105 | O/M | O/M |
| | FPINN | NN'2025 | 0.0331 | 0.0213 | O/M | O/M | O/M | 0.0210 | 0.0136 | O/M | O/M | O/M |
| | CoPINN | Ours | **0.0016** | **0.0006** | **0.0004** | **0.0003** | **0.0002** | **0.0010** | **0.0005** | **0.0002** | **0.0002** | **0.0002** |
| | IMP. | - | 74% | 70% | 69% | 63% | 78% | 74% | 62% | 62% | 60% | 67% |
| (3+1)-d Klein-Gordon | PINN | JCP'2019 | 0.0129 | O/M | O/M | O/M | O/M | 0.0096 | O/M | O/M | O/M | O/M |
| | gPINN | CMAME'2022 | 0.0144 | O/M | O/M | O/M | O/M | 0.0107 | O/M | O/M | O/M | O/M |
| | SPINN | NeurIPS'2023 | 0.0151 | 0.0086 | 0.0073 | O/M | O/M | 0.0112 | 0.0064 | 0.0054 | O/M | O/M |
| | SPINN (m) | NeurIPS'2023 | 0.0078 | 0.0065 | 0.0077 | O/M | O/M | 0.0058 | 0.0048 | 0.0057 | O/M | O/M |
| | AHD-PINN | IJCAI'2024 | 0.0109 | 0.0107 | O/M | O/M | O/M | 0.0081 | 0.0080 | O/M | O/M | O/M |
| | RoPINN | NeurIPS'2024 | 0.0135 | O/M | O/M | O/M | O/M | 0.0100 | O/M | O/M | O/M | O/M |
| | FPINN | NN'2025 | 0.0152 | O/M | O/M | O/M | O/M | 0.0113 | O/M | O/M | O/M | O/M |
| | CoPINN | Ours | **0.0041** | **0.0027** | **0.0017** | **O/M** | **O/M** | **0.0031** | **0.0019** | **0.0012** | **O/M** | **O/M** |
| | IMP. | - | 47% | 58% | 78% | / | / | 47% | 60% | 79% | / | / |
| Diffusion | PINN | JCP'2019 | 0.0095 | 0.0082 | 0.0081 | O/M | O/M | 0.00082 | 0.00071 | 0.00070 | O/M | O/M |
| | gPINN | CMAME'2022 | 0.0099 | 0.0087 | 0.0090 | O/M | O/M | 0.00086 | 0.00076 | 0.00078 | O/M | O/M |
| | SPINN | NeurIPS'2023 | 0.0447 | 0.0115 | 0.0075 | 0.0061 | 0.0061 | 0.00387 | 0.00100 | 0.00065 | 0.00053 | 0.00053 |
| | SPINN (m) | NeurIPS'2023 | 0.0390 | 0.0067 | 0.0041 | 0.0036 | 0.0036 | 0.00338 | 0.00058 | 0.00036 | 0.00031 | 0.00031 |
| | AHD-PINN | IJCAI'2024 | 0.0102 | 0.0080 | 0.0074 | O/M | O/M | 0.00088 | 0.00069 | 0.00064 | O/M | O/M |
| | RoPINN | NeurIPS'2024 | 0.0412 | 0.0308 | 0.0306 | O/M | O/M | 0.00357 | 0.00267 | 0.00265 | O/M | O/M |
| | FPINN | NN'2025 | **0.0058** | 0.0048 | O/M | O/M | O/M | 0.00050 | 0.00042 | O/M | O/M | O/M |
| | CoPINN | Ours | **0.0058** | **0.0038** | **0.0035** | **0.0033** | **0.0033** | **0.00047** | **0.00033** | **0.00030** | **0.00028** | **0.00028** |
| | IMP. | - | 0% | 21% | 12% | 8% | 8% | 6% | 21% | 17% | 10% | 10% |

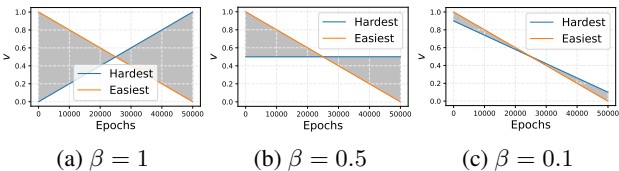

(a) $\beta = 1$      (b) $\beta = 0.5$      (c) $\beta = 0.1$

*Figure 3.* Different $\beta$ values in cognitive training scheduler result in a range of changes in weight $v$.

## 3. Experiments

### 3.1. Experimental Setup

**Datasets**. To show the performance of solving PDEs, we carry out experiments on six popular public datasets, including 1D Convection Equation, 3D (i.e., Diffusion Equation, Helmholtz Equation, (2+1)-d Klein-Gordon Equation, and Flow Mixing Problem) and 4D (i.e., (3+1)-d Klein-Gordon Equation) PDE systems. During training, we perform all experiments on different numbers of collocation points $N_c$, i.e., $16^3$, $32^3$, $64^3$, $128^3$, and $256^3$. Due to space limitations, the experimental results of 1D Convection Equation and (2+1)-d Flow Mixing Problem are shown in the Appendix C.4 and Appendix C.5.

**Baselines.** To verify the effectiveness of our method, we compare the proposed CoPINN with seven state-of-the-art PINN methods, including: PINN (Raissi et al., 2019), gPINN (Yu et al., 2022), SPINN (Cho et al., 2023), SPINN(m) (Cho et al., 2023), AHD-PINN (Dashtbayaz et al., 2024), RoPINN (Wu et al., 2024), and FPINN (Wu et al., 2025).

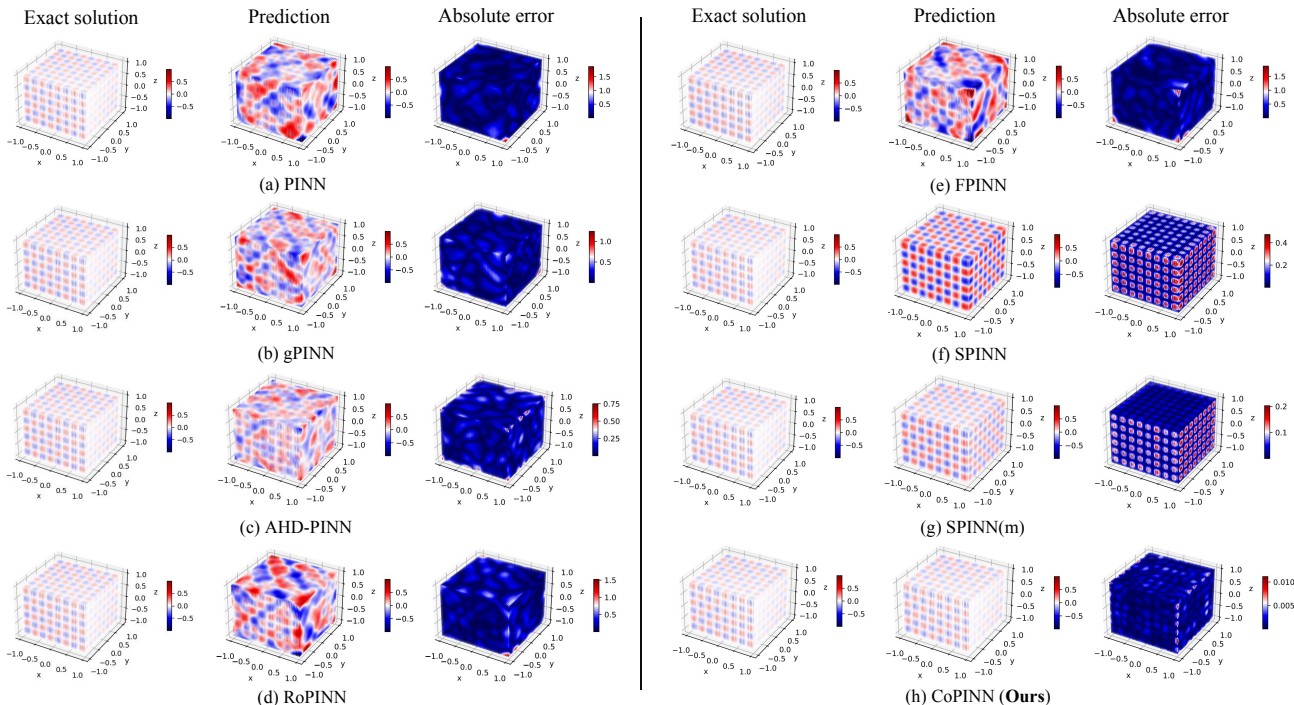

*Figure 4.* Prediction results of CoPINN and the seven baselines on the Helmholtz dataset with $N_c = 32^3$. The exact solution, prediction, and absolute error are shown on the left, middle, and right, respectively.

**Evaluation Metrics**. To comprehensively evaluate our CoPINN, we employ two evaluation metrics: Relative $L_2$ error ($RL_2$) and Root Mean Square Error ($RMSE$). To highlight the advantages of our method, we report the performance improvement (IMP.).

More details of datasets, baselines, evaluation metrics, and the details of implementation are summarized in the Appendix B.1, Appendix B.2, Appendix B.3, and Appendix B.4, respectively.

### 3.2. Comparison with State-of-the-Art Methods

To comprehensively evaluate the performance of our CoPINN, we compare it with seven state-of-the-art baselines. The experimental results on various numbers of collocation points are shown in Table 1, from which we can observe that:

- As the number of collocation points increases, the $RL_2$ and $RMSE$ metrics of most methods gradually decrease. From these results, we can improve the prediction precision by increasing the number of collocation points. In the cases of multiple collocation points (i.e., $N_c = 128^3$ and $N_c = 256^3$), some comparison methods (such as PINN, gPINN, AHD-PINN, RoPINN, and FPINN) fail to complete training due to out-of-memory (denoted as O/M). In contrast, by leveraging its separable architecture, CoPINN mitigates the burden of

representing the entire 3D function and completes training.

- Under any number of collocation points, CoPINN consistently achieves the best $RL_2$ and $RMSE$ metrics across all datasets. It is worth noting that on the Helmholtz equation with $N_c = 256^3$, CoPINN achieves an $RL_2$ of just 0.0006. This represents a 98% improvement compared to the second-best method, i.e., SPINN(m). These results validate the effectiveness of our easy-to-hard cognitive training scheduler.

- For the Diffusion equation, the $RL_2$ of almost all the comparison methods can achieve the order of 1e-3. This suggests that the Diffusion equation is relatively straightforward to learn. In contrast, on the other three equations (i.e., Helmholtz, (2+1)-d and (3+1)-d Klein-Gordon), the $RL_2$ of most comparison methods typically only reaches the order of 1e-1 or 1e-2. This discrepancy arises because these three equations feature numerous mutation regions, making them more challenging to learn. Our method learns data from easy to hard through the cognitive training scheduler, which allows the model to gradually optimize, thereby avoiding local optimality and achieving higher IMP for more challenging PDEs. This further illustrates the effectiveness of CoPINN in learning hard PDEs.

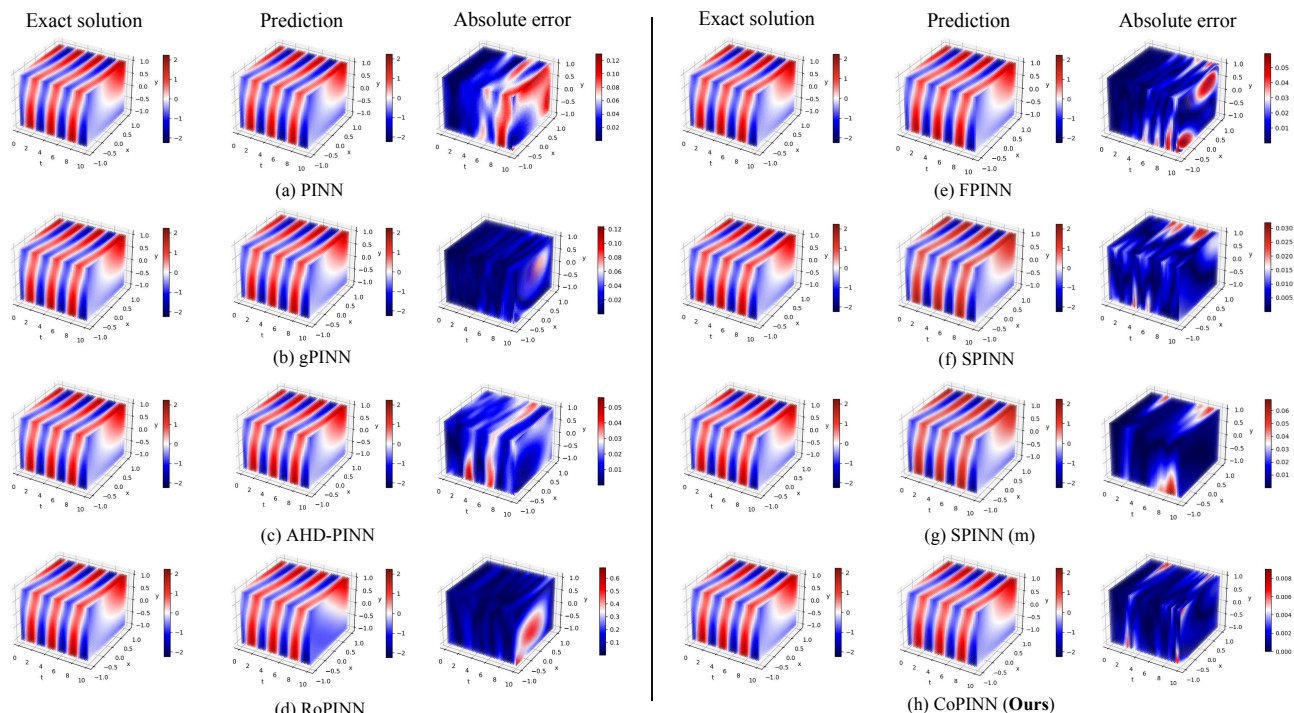

*Figure 5.* Prediction results of our proposed method and the state-of-the-art baselines method on the (2+1)-d Klein-Gordon dataset with $N_c = 32^3$. The exact solution, prediction, and absolute error are shown on the left, middle, and right, respectively.

*Table 2.* Ablation study on the (2+1)-d Klein-Gordon equation. Evaluation metric is $RL_2$.

| Methods \ $N_c$ | $16^3$ | $32^3$ | $64^3$ | $128^3$ | $256^3$ |
|---|---|---|---|---|---|
| CoPINN-1 | 0.0062 | 0.0020 | 0.0013 | 0.0008 | 0.0009 |
| CoPINN-2 | 0.0037 | 0.0027 | 0.0015 | 0.0013 | 0.0011 |
| CoPINN-3 | 0.0016 | 0.0010 | 0.0008 | 0.0008 | 0.0005 |
| CoPINN (Ours) | **0.0016** | **0.0006** | **0.0004** | **0.0003** | **0.0002** |

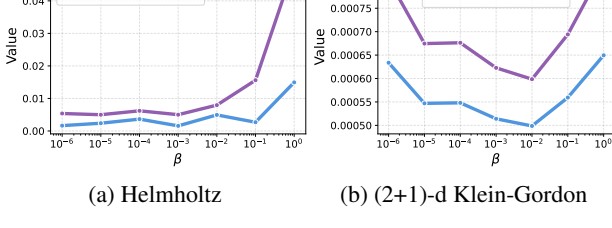

(a) Helmholtz      (b) (2+1)-d Klein-Gordon

*Figure 6.* Performance of CoPINN with varying $\beta$ on the Helmholtz equation and the (2+1)-d Klein-Gordon equation with $N_c = 32^3$.

### 3.3. Ablation Study

The proposed cognitive training scheduler in our CoPINN consists of two components, i.e., the epoch change component ($\tau_e \cdot (i-1)$), and the sample change component ($\beta \cdot \delta_j^i$). To demonstrate the effectiveness of each component, we conduct an ablation study on the (2+1)-d Klein-Gordon equation with collocation point number $N_c = 32^3$. To be specific, we design the following three variations of CoPINN, i.e., CoPINN-1, CoPINN-2, and CoPINN-3. Among them, CoPINN-1 represents using the original loss function, i.e., Equation (4). CoPINN-2 represents removing the epoch change component in Equation (11), i.e., the weights are calculated by $v_j^i = v_e^1 - \beta \cdot \delta_j^i$. CoPINN-3 represents removing the sample change component in Equation (11), i.e., the weights are calculated by $v_j^i = v_e^1 - \tau_e \cdot (i-1)$. The

results are shown in Table 2. We can draw the following observations: **(1)** Both of these two components are essential, and missing either will result in performance degradation. **(2)** The epoch change component is more important than the sample change component. More detailed ablation study is provided in Appendix C.1.

### 3.4. Visualisation Analysis

To more comprehensively compare our CoPINN with baselines, we present the exact solution, prediction results, and absolute errors for each method on the Helmholtz and (2+1)-d Klein-Gordon equation with collocation points $N_c = 32^3$. The visualization results are shown in Figure 4 and Figure 5.

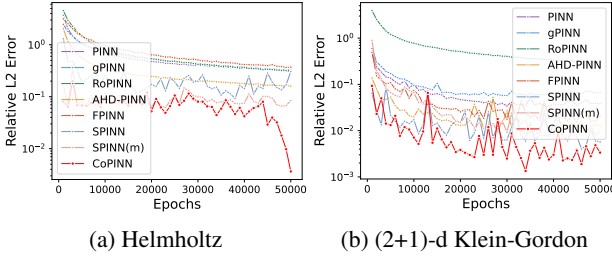

(a) Helmholtz      (b) (2+1)-d Klein-Gordon

*Figure 7.* Variation curve of $RL_2$ with the increase of epoch under $N_c = 32^3$.

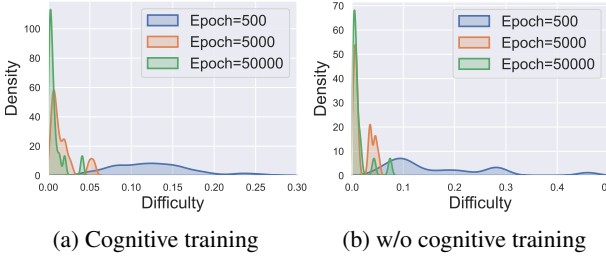

(a) Cognitive training      (b) w/o cognitive training

*Figure 8.* Difficulty density on different epochs. Experiments are conducted on the (3+1)-d Klein Gordon equation PDE systems with $16^3$ collocation points.

We can observe that: **(1)** All comparison methods exhibit a significantly poorer fitting performance on the Helmholtz equation than on the (2+1)-dimensional Klein-Gordon equation. This suggests that the Helmholtz equation is more challenging to learn. **(2)** CoPINN closely aligns with the exact solutions for both the more difficult Helmholtz equation and the relatively easier (2+1)-dimensional Klein-Gordon equation, especially in difficult regions such as physical boundaries, which is attributed to its progressive optimization of the entire sampling regions from easy to hard. More visualisation and analysis on the Diffusion equation are shown in Appendix C.2.

### 3.5. Parametric Analysis

To investigate the impact of different $\beta$-hyperparameter values in Equation (11) on the performance of CoPINN, we plot the performance curves of CoPINN with varying $\beta$ on the Helmholtz equation and the (2+1)-d Klein-Gordon equation with $N_c = 32^3$ in Figure 6. On the Helmholtz equation, the performance of CoPINN initially remains stable as $\beta$ increases, but then rapidly declines (with a sharp rise in error). For the (2+1)-d Klein-Gordon equation, the performance of CoPINN first improves and then deteriorates as $\beta$ increases, achieving optimal performance when $\beta = 10^{-2}$. Therefore, we recommend that the value of $\beta$ be chosen within $[10^{-4}, 10^{-1}]$.

### 3.6. Error Analysis

To further observe the performance trend of our CoPINN as the number of iterations increases, we plot the $RL_2$ variation curves for each method across epochs on the Helmholtz and (2+1)-d Klein-Gordon equations. As shown in Figure 7, we can observe that: **(1)** Although the $RL_2$ curves exhibit some fluctuations, CoPINN shows an overall downward trend. This shows that CoPINN is correctly optimized. **(2)** CoPINN outperforms all comparison methods for the majority of the epochs, and as the number of epochs increases, its final performance consistently surpasses that of the others. This improvement is attributed to CoPINN gradually learning the entire sampling region from easy to difficult, thus overcoming the UPP. Besides, results and analysis on the RMSE metric can be found in Appendix C.3.

### 3.7. Difficulty Analysis

To evaluate the effectiveness of the cognitive training scheduler of CoPINN, we visualize the difficulty density of samples under different iterations. The results are shown in the Figure 8, from which we can observe that: **(1)** With the increase in the number of iterations, the overall difficulty gradually decreases with or without the cognitive training scheduler. **(2)** Compared with without cognitive training scheduler, our method can reduce the difficulty of the samples, thanks to gradually increasing the weight of the difficult samples during the training phase.

## 4. Conclusion

In this paper, we reveal and address a previously unexplored yet pervasive challenge in PINN, referred to as the Unbalanced Prediction Problem (UPP). To tackle this issue, we propose a novel Cognitive Physics-Informed Neural Networks (CoPINN) method, which imitates the human cognitive learning manner from easy to hard. Specifically, our CoPINN first adopt separable learning to encode each independent one-dimensional coordinate by each separable sub-network, thereby reducing computational complexity. Then, an aggregation scheme is applied to produce multi-dimensional predictions. During the training process, CoPINN dynamically evaluates the difficulty of each sample based on the gradient magnitude of the PDE residuals. Finally, we propose a cognitive training scheduler that adaptively optimizes the PINN model from easy to difficult samples, improving its robustness and generalization, especially in predicting physical boundary regions. To demonstrate its versatility, we evaluate CoPINN on multiple widely used PDEs. Experimental results show that CoPINN consistently outperforms all comparison methods.

## Acknowledgements

This work was supported by the Open Research Project of the National Key Laboratory of Fundamental Algorithms and Models for Engineering Numerical Simulation.

## Impact Statement

This paper presents work whose goal is to advance the field of Machine Learning. There are many potential societal consequences of our work, none of which we feel must be specifically highlighted here.

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

# APPENDIX

This document provides additional details, analysis, and experimental results to support the main submission.

## A. CoPINN Training Algorithm

Algorithm 1 describes the training algorithm of Cognitive Physics-Informed Neural Networks (CoPINN). First, the separable sub-networks are used to encode independent one-dimensional coordinates. Second, an aggregation scheme is applied to obtain the multi-dimensional predicted physical variables. During the training process, CoPINN dynamically evaluates the difficulty of predicting each sample based on the gradient magnitude of the PDE residuals. Finally, a cognitive training scheduler is employed to adaptively optimize the model from easy to hard.

---
**Algorithm 1** CoPINN algorithm
---

**Input:** maximal epoch number $N_e$, initial parameters of sub-networks of CoPINN $\{f_j^{(\theta_j)}\}_{j=1}^r$, training set $\mathcal{D}$, hyperparameter $\beta$.
**Output:** optimized network parameters $\{\theta_j\}_{j=1}^r$.
Compute epoch step size $\tau_e$ by Equation (7).
**for** $i = 1$ to $N_e$ **do**
    Calculate the output of each sub-network and merge them by Equation (2).
    Calculate original PDE loss $\mathcal{L}_{pde}$, IC loss $\mathcal{L}_{ic}$, and BC loss $\mathcal{L}_{bc}$.
    Compute the weight assigned to the easiest sample point in epoch $i$, i.e., $v_e^i$, by Equation (8).
    Compute the weight assigned to the hardest sample point in epoch $i$, i.e., $v_h^i$, by Equation (9).
    Compute difficulty $D_k^i$ for each sample point $\mathbf{x}_k^i$ by Equation (6).
    Compute weights for each sample point according to their difficulty by Equation (11).
    Compute weighted loss by Equation (5).
    Update $\{\theta_j\}_{j=1}^r$.
**end for**

---

## B. Detailed Experimental Setup

### B.1. Dataset Details

Details of the five partial differential equations used in the experiment are as follows.

**Helmholtz equation.** The Helmholtz equation is a second-order partial differential equation that is commonly used to describe phenomena such as waves and vibrations. It has important applications in physics, engineering, acoustics, electromagnetism, and quantum mechanics. The form of the Helmholtz equation we selected is as follows:

$$\Delta u + k^2 u = q, \quad x \in \Omega, \tag{12}$$

$$u(x) = 0, \quad x \in \partial\Omega, \tag{13}$$

where $\Omega$ is the spatial domain, $\Omega = [-1, 1]^3$, $q$ is the given source term, $q = -(a_1\pi)^2 u - (a_2\pi)^2 u - (a_3\pi)^2 u + k^2 u$, $u$ is the manufactured solution, $u = \sin(a_1\pi x_1)\sin(a_2\pi x_2)\sin(a_3\pi x_3)$, where $k = 1, a_1 = 4, a_2 = 4, a_3 = 3$.

**Diffusion Equation.** Diffusion equations are mathematical models used to describe changes in the diffusion or distribution of matter, energy, or other physical quantities in space. It has a wide range of applications in physics, chemistry, biology, and engineering, such as heat conduction, pollutant diffusion, gas diffusion, and so on. The form of the Diffusion equation we selected is as follows:

$$u_t = \alpha(\|\nabla u\|^2 + u\Delta u), \quad x \in \Omega, t \in \Gamma, \tag{14}$$

$$u(x, 0) = u_{ic}(x), \quad x \in \Omega, \tag{15}$$

$$u(x, t) = 0, \quad x \in \partial\Omega, t \in \Gamma, \tag{16}$$

where $\alpha$ is the diffusion coefficient, $\alpha = 0.05$, $\Omega$ is the spatial domain, $\Omega = [-1, 1]^2$, $\Gamma$ is the temporary domain, $\Gamma = [0, 1]$. $u_{ic}$ is the initial condition. Following SPINN (Cho et al., 2023), we use the PDE simulation software FEniCS to obtain the numerical solution as the reference, with a resolution of 101×101×101. The initial condition $u_{ic}$ is a superposition of three Gaussian functions, as shown below:

$$
\begin{aligned}
u_{ic}(x, y) = &\ 0.25 \exp[-10\{(x - 0.2)^2 + (y - 0.3)^2\}] \\
&+ 0.4 \exp[-15\{(x + 0.1)^2 + (y + 0.5)^2\}] \\
&+ 0.3 \exp[-20\{(x + 0.5)^2 + y^2\}].
\end{aligned}
\tag{17}
$$

**Klein-Gordon Equation.** The Klein-Gordon equation is an equation in quantum field theory that describes scalar fields. It is an important equation for describing particles in relativistic quantum field theory and is a wave equation under the framework of special relativity, especially for describing massless spin particles (such as scalar particles). The solution of this equation can be used to describe the quantum states of some quantum fields and can be generalized to describe particles with spin (such as electrons and photons). The form of the Klein-Gordon equation we selected is as follows:

$$u_{tt} - \Delta u + u^2 = f, \quad x \in \Omega, t \in \Gamma, \tag{18}$$

$$u(x, 0) = x_1 + x_2, x \in \Omega, \quad x \in \Omega, \tag{19}$$

$$u(x, t) = u_{bc}(x), \quad x \in \partial\Omega, t \in \Gamma, \tag{20}$$

where $\Omega$ is the spatial domain, $\Omega = [-1, 1]^2$, $\Gamma$ is the temporary domain, $\Gamma = [0, 10]$, $u$ is the manufactured solution, $u = (x_1 + x_2)\cos(2t) + x_1 x_2 \sin(2t)$, $f, u_{bc}$ is extracted from manufactured solution $u$.

**(2+1)-d Flow Mixing Problem**. Flow mixing is a crucial topic in fluid mechanics, typically involving the interaction of multiple fluid components or fluids with varying temperatures and densities during flow. The Flow Mixing equations that govern this process are derived from the laws of conservation of mass, momentum, and energy, along with the transport properties of the fluid, such as diffusion, convection, and turbulence. In this paper, we model flow mixing as a time-dependent partial differential equation that describes the mixing behavior of two fluids at the interface within a two-dimensional environment, as shown in the following form:

$$u(t, x, y) = -\tanh(\frac{y}{2}\cos(\omega t) - \frac{x}{2}\sin(\omega t)), \quad t \in [0, 4], x \in [-4, 4], y \in [-4, 4], \tag{21}$$

$$\omega = \frac{1}{r}\frac{v_t}{v_{t,max}} \tag{22}$$

$$u_t + \alpha u_x + \beta u_y = 0, \tag{23}$$

$$\alpha(x,y) = -\frac{v_t}{v_{t,max}}\frac{y}{r}, \tag{24}$$

$$\beta(x,y) = -\frac{v_t}{v_{t,max}}\frac{x}{r}, \tag{25}$$

$$v_t = \text{sech}^2(r)\tanh(r), \tag{26}$$

$$r = \sqrt{x^2 + y^2}, \tag{27}$$

$$v_{t,max} = 0.385. \tag{28}$$

**Convection Equation**. In this study, we consider a 1D Convection equation, which is commonly used to model transport phenomena, described as follows:

$$\frac{\partial u}{\partial t} + \xi\frac{\partial u}{\partial x} = 0, \quad x \in [0, 2\pi], \quad t \in [0, 1] \tag{29}$$
$$u(x,0) = h(x) \tag{30}$$
$$u(0,t) = u(2\pi, t) \tag{31}$$

where $\xi$ is the convection coefficient and $h(x)$ is the initial condition. Following (Krishnapriyan et al., 2021), we use a constant setting of $h(x) = sin(x)$ with periodic boundary conditions in all our experiments, while varying the value of $\xi$ in different case studies.

### B.2. Evaluation Baselines

We compare the proposed CoPINN with the following methods:

- **PINN** (Physics-informed neural networks) (Raissi et al., 2019) is the first attempt to integrate physical information, namely, the constraints of PDEs, into neural network optimization. This integration allows the model to learn the data distribution during training while adhering to physical laws, thereby enhancing both its generalization ability and physical consistency.

- **gPINN** (Gradient-enhanced physics-informed neural networks) (Yu et al., 2022) leverages gradient information of the PDE residual and embeds the gradient into the loss function, thereby improving the accuracy and training efficiency of PINN.

- **SPINN** (Separable physics-informed neural networks) (Cho et al., 2023) propose a new architecture to operate on a per-axis basis, thereby significantly reducing the number of network propagations in multi-dimensional PDEs. Based on SPINN, the author also proposed SPINN with modified MLP (**SPINN (m)**), which uses modified MLP to replace MLP and achieve better performance.

- **AHD-PINN** (PINN with Higher Derivative of the Activation function satisfies bijective) (Dashtbayaz et al., 2024) demonstrates the global minimization of residual loss in PINN with sufficient network width, and establishes optimal activation function selection based on bijective derivatives for $k$-th order operators.

- **RoPINN** (Region optimized physics-informed neural networks) (Wu et al., 2024) extends the optimization process of PINN from isolated points to their continuous neighborhood regions, which can theoretically decrease the generalization error, especially for hidden high-order constraints of PDEs.

- **FPINN** (Deep fuzzy physics-informed neural networks) (Wu et al., 2025) is the first attempt to apply fuzzy neural network to solve PDEs, which can handle the ambiguity in data from commercial simulation software.

## B.3. Evaluation Metrics

To comprehensively evaluate our CoPINN, we introduce two evaluation metrics: Relative $L_2$ error ($RL_2$) and Root Mean Square Error ($RMSE$). To be specific, for the forward problem of partial differential equations, the Relative $L_2$ error is defined as:

$$RL_2 = \frac{1}{m} \sum_{i=1}^{m} \frac{\left\| u_{pred}^i - u_{exact}^i \right\|_2}{\left\| u_{exact}^i \right\|_2}, \tag{32}$$

where $u_{pred}^i$ and $u_{exact}^i$ indicate the prediction and reference solutions of the $i$-th collocation point, $m$ is the number of collocation points, respectively.

The second evaluation metric is the Root Mean Square Error, given by:

$$RMSE = \sqrt{\frac{1}{m} \sum_{i=1}^{m} (u_{pred}^i - u_{exact}^i)^2}. \tag{33}$$

The calculation methods for $RL_2$ and $RMSE$ are referenced in the original PINN paper (Raissi et al., 2019) and the SPINN paper (Cho et al., 2023).

The third evaluation metric is the improvement percentage (IMP.), given by:

$$\text{IMP.} = \frac{|u_s - u_b|}{u_s} \times 100\%, \tag{34}$$

where $u_s$ represents the $RL_2$ or $RMSE$ of the second-best method, while $u_b$ denotes the $RL_2$ or $RMSE$ of the best method.

## B.4. Implementation Details

The results of all the experiments were averaged from five random seeds. For Helmholtz, (2+1)-d Klein-Gordon, (3+1)-d Klein-Gordon, Diffusion and flow Mixing 3D equations, we set the sampling points to $100^3$, $100^3$, $50^3$, $101^3$, and $100^3$ during testing, respectively. Following SPINN (Cho et al., 2023), we divide the dataset into a training set and a test set. The training set is used to train the neural network, while the test set is used to evaluate the prediction ability of the model. For all comparative methods, we use the source codes provided by the original authors and apply the parameters recommended in their published papers. For our CoPINN, the network architecture consists of five hidden layers, where each layer has 128 hidden units. We apply the modified MLP introduced in (Wang et al., 2021a) to CoINN. On all datasets, we exploit the Adam optimizer (Kingma & Ba, 2014) to train our model with a large learning rate of 1e-3 for 50,000 epochs. We use the 'tanh' activation function throughout our CoPINN. According to the parameter analysis of our method, on the (2+1)-d Klein-Gordon dataset, we set $\beta = 0.01$. And on the Helmholtz, Diffusion, and (3+1)-d Klei-Gordon datasets, $\beta$ is set to 0.001. To ensure a fair comparison, we set the balance parameter of the loss terms to be equal, i.e., $\lambda_* = 1$ in Equation (5). All experiments are implemented in JAX/Flax and trained on a single NVIDIA 3090 GPU with 24GB of memory.

# C. Additional Experimental Results

## C.1. Additional Ablation Study

This section serves as a supplement to the ablation study in Section 3.3. To demonstrate the effectiveness of each component in the cognitive training scheduler of our CoPINN, we perform an ablation study on the (2+1)-d Klein-Gordon equation using the RMSE metric. Specifically, we design the following three variations of CoPINN, i.e., CoPINN-1, CoPINN-2, and CoPINN-3. Specifically, CoPINN-1 uses the original loss function, i.e., Equation (4). CoPINN-2 represents removing the epoch change component in Equation (11), i.e., the weights are calculated by $v_j^i = v_e^1 - \beta \cdot \delta_j^i$. CoPINN-3 represents removing the sample change component in Equation (11), i.e., the weights are calculated by $v_j^i = v_e^1 - \tau_e \cdot (i - 1)$. The results are shown in Table 3 demonstrate that: **(1)** Both of these two components are essential, and missing either will result in performance degradation. **(2)** The epoch change component is more important than the sample change component. These observations are the same as leveraging $RL_2$.

*Table 3.* Ablation study on the (2+1)-d Klein-Gordon equation. Evaluation metric is $RMSE$.

| Methods \ $N_c$ | $16^3$ | $32^3$ | $64^3$ | $128^3$ | $256^3$ |
|---|---|---|---|---|---|
| CoPINN-1 | 0.0039 | 0.0013 | 0.0008 | 0.0005 | 0.0002 |
| CoPINN-2 | 0.0023 | 0.0017 | 0.0010 | 0.0008 | 0.0007 |
| CoPINN-3 | 0.0010 | 0.0006 | 0.0005 | 0.0005 | 0.0003 |
| **CoPINN (Ours)** | **0.0010** | **0.0005** | **0.0002** | **0.0002** | **0.0002** |

## C.2. Additional Visualisation Analysis

This section serves as a supplement to the visualisation analysis presented in Section 3.4. To more comprehensively compare CoPINN with baselines, we present the exact solution, prediction, and absolute error for each method on the Diffusion equation in Figure 9. The selected time points are $t = 0, 0.5, 1$. The prediction results of our CoPINN are highly consistent with the exact solution, further confirming its superior performance.

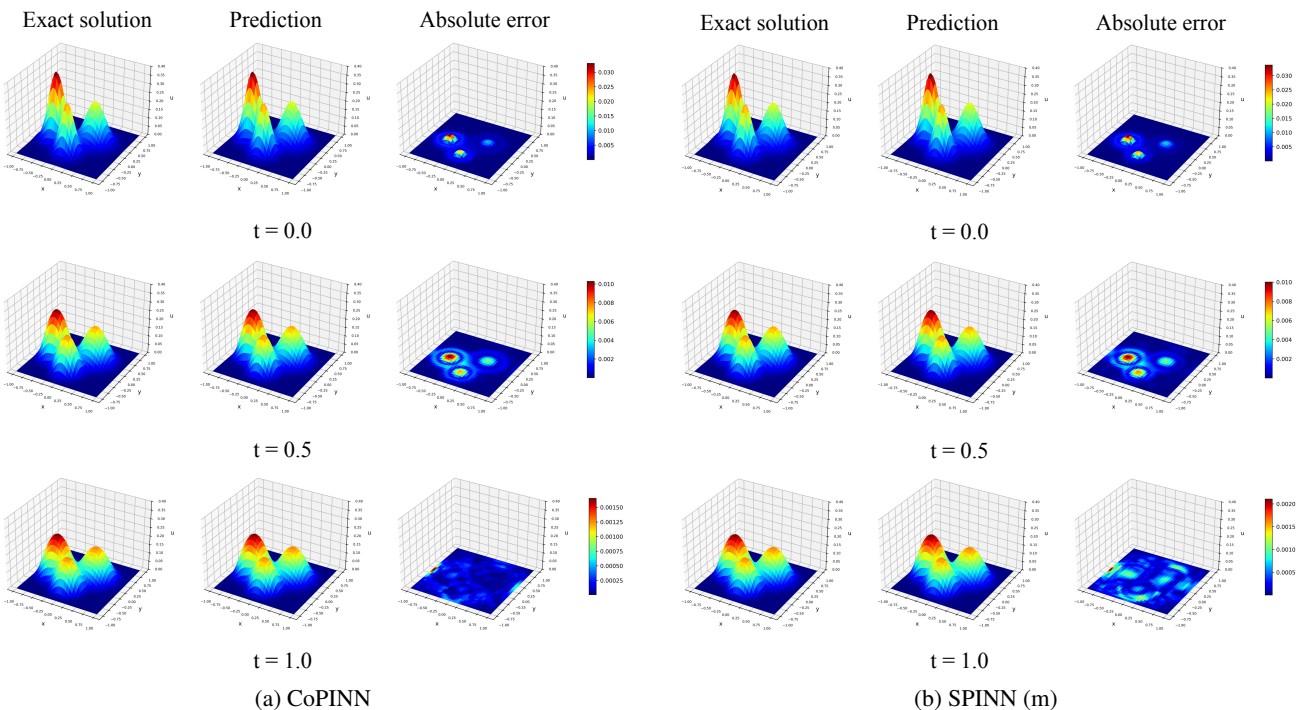

*Figure 9.* Prediction results of CoPINN and SPINN (m) on the Diffusion dataset with $N_c = 256^3$. The left is the exact solution, the middle is the predicted value, and the right is the absolute value of the error.

## C.3. Additional Error Analysis

This section serves as a supplement to the error analysis presented in Section 3.6. To further observe the performance trend of our CoPINN as the number of iterations increases, we plot the $RMSE$ metric variation curves for each method across epochs on the Helmholtz and (2+1)-d Klein-Gordon equations. As shown in Figure 10, similar observations can be obtained as in the $RL_2$ metric: despite fluctuations, $RMSE$ curves for all methods generally decrease over iterations, showing effective optimization, with CoPINN consistently outperforming baselines and achieving superior final performance as epochs increase. These consistent improvement underscores the superiority and stability of our CoPINN.

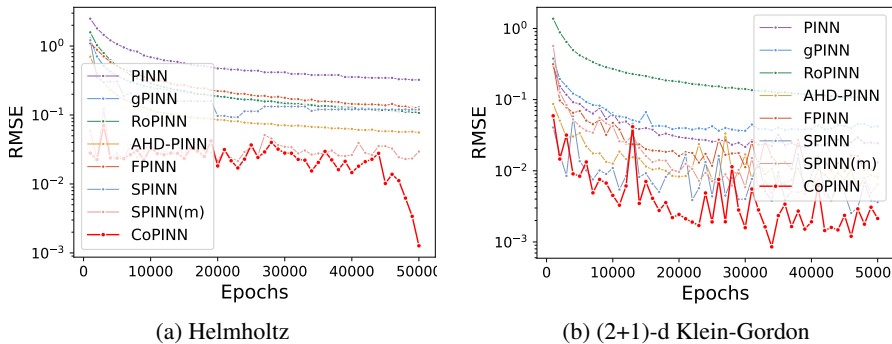

(a) Helmholtz                                   (b) (2+1)-d Klein-Gordon

*Figure 10.* Variation curve of $RMSE$ with the increase of epoch under $N_c = 32^3$.

## C.4. Experiments on Flow-mixing 3D Equation

To further validate the performance advantages of the proposed CoPINN, we compare the results of CoPINN, SPINN, and SPINN(m) on a more complex Flow-mixing 3D Equation. As demonstrated by the results in Table 4, CoPINN achieves the smallest error on the Flow-mixing 3D Equation, further confirming its superiority.

*Table 4.* Full results of the Flow-mixing 3D equation. $N_c$ is the number of collocation points. The best and the second-best results are highlighted in **boldface** and underlined, respectively.

| Evaluation Metric | | $RL_2$ | | | | | $RMSE$ | | | | |
|---|---|---|---|---|---|---|---|---|---|---|---|
| Methods | $N_c$ 
 Ref. | $16^3$ | $32^3$ | $64^3$ | $128^3$ | $256^3$ | $16^3$ | $32^3$ | $64^3$ | $128^3$ | $256^3$ |
| SPINN | NeurIPS'2023 | 0.1352 | 0.1054 | 0.0861 | 0.0787 | 0.0672 | 0.1087 | 0.0875 | 0.0711 | 0.0575 | 0.0488 |
| SPINN (m) | NeurIPS'2023 | 0.0884 | 0.0263 | 0.0080 | **0.0032** | 0.0023 | 0.0342 | 0.0241 | 0.0063 | **0.0022** | 0.0016 |
| CoPINN | Ours | **0.0853** | **0.0230** | **0.0063** | **0.0032** | **0.0022** | **0.0323** | **0.0190** | **0.0060** | **0.0022** | **0.0015** |
| IMP. | - | 4% | 11% | 21% | 0% | 4% | 6% | 21% | 5% | 0% | 6% |

## C.5. Experiments on Convection Equation

To further validate the effectiveness of the proposed difficulty evaluation and cognitive training scheduler on the 1D Convection equation, we employ them on the vanilla PINN. In this experiment, we set the number of collocation points to 1000, set the $\beta$ in the cognitive training scheduler as 0.1, do not change other default parameters, and average across 3 random runs. The experimental results below show that, in Equation (29), when $\xi = 30$, our method outperforms all the comparison methods, and when $\xi = 50$, our method is also competitive compared to R3 and Casual R3.

*Table 5.* Relative L2 errors of comparative methods over 1D Convection equation.

| Methods $\xi$ | 30 | 50 |
|---|---|---|
| RAD (Wu et al., 2023) | 0.0418 | 0.6547 |
| R3 (Daw et al., 2023) | 0.0160 | 0.5794 |
| Causal R3 (Daw et al., 2023) | 0.0073 | **0.5471** |
| Ours | **0.0055** | 0.5811 |

# D. Related Work

Physics-Informed Neural Networks (PINN) (Raissi et al., 2019) is a machine learning model that integrates physical laws into the training process of neural networks. Its goal is to solve partial differential equations (PDEs) and other complex scientific problems by combining data-driven learning with known physical principles. Due to its meshless character and low computational cost, PINN has become a promising alternative to traditional numerical methods. Thus, in recent years, a large number of PINN methods have been proposed (Song & Alkhalifah, 2021; Cai et al., 2021; Wang et al., 2022; Jin et al., 2024; Zou et al., 2024b). With the fast development of deep learning, the PINN technology has been widely explored from various aspects: novel neural network architectures, novel optimization schemes, loss-weighting, and adaptive sampling. In the following, we review these four aspects of technology.

**Neural network architectures** attempt to design novel neural network architectures to enhance model capacity. For example, cPINN (Jagtap et al., 2020) constructs a separate neural network in each discrete sub-domain. However, cPINN struggles with solving PDE problems involving noisy data. To address this limitation, BPINN (Yang et al., 2021) introduces a Bayesian framework to quantify aleatoric uncertainty caused by noisy data. Despite their effectiveness, BPINN incurs significant time overhead. To reduce computational costs while handling noisy data, FPINN (Wu et al., 2025) leverages fuzzy membership functions and fuzzy rules to manage noise and uncertainty. Nevertheless, these methods face challenges in solving high-dimensional PDEs or approximating highly complex solution functions due to their high computational and memory demands. To tackle these issues, SPINN (Cho et al., 2023) adopts a per-axis operation approach, significantly reducing the network propagations required for multi-dimensional PDEs.

**Optimization schemes** aim to develop new optimization schemes to cope with the rough loss landscape. For instance, hp-vPINN (Kharazmi et al., 2021) solves PDEs by minimizing residuals through a least-squares approach. gPINN (Yu et al., 2022) incorporates gradient information from PDE residuals into the loss function to enhance precision and training efficiency. However, these methods often violate the temporal causality property, leading to training challenges for evolutionary PDEs. To address this issue, TL-DPINN (Li et al., 2024) adopts implicit time differencing to preserve temporal causality and utilize transfer learning to update PINN sequentially. Despite their advantages, TL-DPINN optimizes models only on scattered points. To achieve accurate solutions across the entire domain, RoPINN (Wu et al., 2024) extends the optimization process to continuous neighborhood regions, thereby improving the performance of PINN.

**Loss re-weighting** aims to develop novel loss re-weighting to balance different losses, thereby achieving more balanced convergence. To be specific, Wang et al. (2022) observed that the weighted combination of competitive multiple loss functions significantly impacts the training of PINN. To address this, they propose lbPINN, which automatically assigns loss weights based on maximum likelihood estimation. Similarly, Xiang et al. (2022) introduces SA-PINN, employing Gaussian Process regression to construct a continuous map of self-adaptive weights. This method dynamically increases weights as the corresponding losses rise during training, enhancing model performance.

**Adaptive sampling** operates as a dynamic optimization framework that iteratively adjusts the spatial-temporal distribution of training points during neural network training. This methodology strategically allocates computational resources by continuously refining the selection of collocation points based on real-time error metrics. Through residual-driven refinement mechanisms, the algorithm preferentially concentrates sampling density in regions exhibiting elevated PDE residual errors (indicative of model inaccuracies) or steep solution gradients (marking mathematically challenging zones like boundary layers, shock fronts, or material interfaces) (Wu et al., 2023; Daw et al., 2023). This targeted resource allocation enables the model to resolve physically critical features with increased precision while avoiding computational redundancy in smoother, well-predicted domains, thereby ensuring enhanced solution accuracy and training efficiency compared to static sampling approaches.

However, the above four aspects of technology always treat both easy and hard samples equally while ignoring sample importance and learning difficulty. To address this issue, we draw inspiration from human cognitive processes and propose a PINN model with SPL that evaluates sample-level difficulty and guides the neural network to fit samples progressively from easy to hard when solving PDEs.

# E. Limitations

Although the proposed CoPINN method demonstrates significant advantages in solving partial differential equations, we acknowledge several limitations that require further investigation. First, when CoPINN increases the weight of difficult samples during each iteration, it may overly focus on these samples. This could result in overfitting to the difficult samples

while neglecting other samples, particularly if the difficult samples do not adequately represent the overall data distribution. Second, dynamically weighting samples can introduce instability into the training process, particularly when the weights assigned to difficult samples increase significantly. This may cause CoPINN to rely disproportionately on these samples during certain iterations, potentially leading to deviations in gradient updates and a more volatile training process.

