# OpenReview forum: "CoPINN: Cognitive Physics-Informed Neural Networks"
_ICML.cc/2025/Conference — ICML 2025 spotlightposter_

### Official Review · Reviewer_JLZV · 2025-03-06

**Overall Recommendation:** 5

**Summary:**

The paper presents a novel framework called Cognitive Physical Informed Neural Network (CoPINN) to address the Unbalanced Prediction Problem. CoPINN employs separable subnetworks to encode one-dimensional coordinates, aggregates them to predict multi-dimensional variables, and dynamically evaluates sample difficulty based on PDE residual gradients. It progressively optimizes sampling regions from easy to hard using a cognitive training scheduler, significantly reducing prediction errors in challenging areas.

**Claims And Evidence:**

yes, this paper is supported by clear and convincing evidence.

**Essential References Not Discussed:**

No.

**Experimental Designs Or Analyses:**

Yes, I checked the soundness and validity of all experimental designs and analyses in this paper.

**Methods And Evaluation Criteria:**

yes, the proposed method and evaluation criteria make sense.

**Other Comments Or Suggestions:**

No.

**Other Strengths And Weaknesses:**

Strengths:

1.The paper clearly articulates the research motivation with illustrative examples.

2.The paper has sufficient experiments, for example, comparative experiments with seven latest methods on five datasets.

3.The proposed method achieves significant improvements over the SOTA, for example, about 90% improvement on Helmholtz and about 70% improvement on (3+1)-d Klein Gordon Equation.

4.This paper innovatively uses self-paced learning to solve the key unbalanced prediction problem (UPP) in the existing PINNs methods.

5.The proposed method is simple but effective.

Weaknesses:

1.Although the results on the Helmholtz and (2+1)-d Klein-Gordon datasets are significantly better than those on the SOTA method, why is the improvement on the (3+1)-d Klein-Gordon and Diffusion datasets limited? This requires further analysis.

2.The authors compared their method with PINNs and FPINNs, which are not good at solving high-dimensional PDEs, and the experimental setting seems unfair.

See questions for more.

**Questions For Authors:**

1.CoPINN consists of Separable learning and Cognitive Training Scheduler. Are the proposed Cognitive Training Scheduler and Separable learning coupled or can they be separated? That is, can the Cognitive Training Scheduler be used for the vanilla PINN?

2.The paper mentions using different numbers of sampling points, i.e. 16^3, 32^3,...,256^3, to train the neural network. Is the same number of sampling points also used for testing? Or is it another number?

3.Hyper parameter $\beta$ has a great impact on the performance of the algorithm. What value is recommended for solving other equations, such as the NS equation?

4.Although the results on the Helmholtz and (2+1)-d Klein-Gordon datasets are significantly better than those on the SOTA method, why is the improvement on the (3+1)-d Klein-Gordon and Diffusion datasets limited? This requires further analysis.

5.What is the performance removing Cognitive Training Scheduler?

**Relation To Broader Scientific Literature:**

Although this paper only focuses on the field of physics-informed neural networks (PINNs) and uses self-paced learning (SPL) to improve the overall performance of PINNs, the proposed SPL method has important reference value for other fields, such as classification, clustering, and retrieval.

**Theoretical Claims:**

Yes, I checked the theoretical part of self-paced learning in this paper.

---

> ### Author Rebuttal · Authors · 2025-04-01
>
> We appreciate your detailed comments. We believe the following point-to-point response can address all the concerns:
>
> **Q1: Weaknesses (1) and Questions For Authors(4)**
>
> **R1:** Compared with Helmholtz and (2+1)-d Klein-Gordon datasets, (3+1)-d Klein-Gordon and Diffusion datasets are relatively less challenging. Therefore, almost all the methods achieve good results. In fact, because most of these two datasets are easy, which leads to limited mining of difficult samples by our CoPINN, and the performance advantage is relatively unobvious. In fact, CoPINN yields improvements of $50\\%$ (Klein-Gordon in (3+1)-d) and $10\\%$ (diffusion equations).
>
> **Q2: Weaknesses (2)**
>
> **R2:** We claim that PINN and FPINN can solve high-dimensional PDEs. However, as the number of training (collocation) points increases, especially for higher-dimensional or more complex problems, their computational burden becomes more pronounced. To solve this problem, we adopt a separable architecture to efficiently handle high-dimensional PDEs. In addition, we also compare with SPINN (which is good at solving high-dimensional PDEs), so the experimental setting is fair.
>
> **Q3: Questions For Authors(1)**
>
> **R3:** Separable learning and Cognitive Training Scheduler are two independent modules. To be specific, Separable Learning mitigates the computational burden in high-dimensional PDE solutions. Conversely, the Cognitive Training Scheduler assesses sample-level difficulty, enabling neural networks to fit samples from easy to hard when solving PDEs.  Cognitive Training Scheduler is plug-and-play and can be used for the vanilla PINN.
>
> **Q4: Questions For Authors(2)**
>
> **R4:** We apologize for the lack of clarity. Following the setup of SPINN, for Helmholtz, (2+1)-d Klein-Gordon, (3+1)-d Klein-Gordon, Diffusion and flow Mixing 3D equations, we set the sampling points to $100^3, 100^3, 50^3, 101^3$, and $100^3$ during testing, respectively. We will include this in the implementation details of the next version.
>
> **Q5: Questions For Authors(3)**
>
> **R5:** Through parameter sensitivity analysis in Figure 5, we find that the performance is better when the $\beta$ value is $0.01$ to $0.00001$. In fact, if $\beta$ is fixed to $0.001$, CoPINN achieves SOTA on all datasets.
> Therefore, for other PDEs, such as the N-S equation, we recommend that the $\beta$ value be selected in $\\{0.01, 0.001, 0.0001, 0.00001\\}$.
>
> **Q6: Questions For Authors(5)**
>
> **R6:** In Section 3.3 of our original paper, we conduct an ablation study to analyze the effectiveness of each component in the cognitive training scheduler of our CoPINN. CoPINN-1 represents using the original loss function, i.e., removing the Cognitive Training Scheduler. The results in Table 2 show that the performance of CoPINN-1 is lower than our proposed CoPINN.

---

> > ### Comment · Reviewer_JLZV · 2025-04-03
> >
> > Thanks for your responses. The authors have overcome all my concerns. This paper reveals and studies a less-touched unbalanced prediction problem (UPP) in PINNs.  By imitating the human cognitive learning process, the authors proposed a novel cognitive PINN framework to adaptively optimize the model from easy to hard, thereby alleviating the negative effect of the hard samples in stubborn regions during the learning process.
> >
> > In general, the motivation of this paper is clear, and the idea is interesting and novel. This approach inspires us to solve the unbalanced prediction problem of stubborn regions in pinn from a cognitive learning perspective. Numerous experiments also show that the proposed method has achieved significant improvement compared with these state-of-the-art methods, which proves the feasibility of cognitive learning for PINN. Therefore, I raise my rate and recommend acceptance of the paper.

---

> > > ### Author Response · Authors · 2025-04-08
> > >
> > > Thank you for your thoughtful review of our manuscript. We sincerely appreciate the time and effort you dedicated to evaluating our work and providing insightful comments.

---

### Official Review · Reviewer_wy2g · 2025-03-11

**Overall Recommendation:** 3

**Summary:**

The paper proposes an adaptive sample weighting strategy for physics-informed neural networks. As a measure of difficulty for a sampling point, the magnitude of the (input) gradient of the PDE residual is proposed. The authors suggest to train PINNs via assigning high sample weights to easy samples early on in the training process and then gradually shift towards hard samples, reducing the weight for the easy ones. The authors then compare their proposed method on a number of example equations and benchmark against several baselines. The experimental results suggest improved accuracy compared to the baseline method.

## update after rebuttal
The authors were responsive and could clear some of my doubts and misunderstandings. I will raise my score.

**Claims And Evidence:**

The improved accuracy and training is illustrated by a reasonable set of numerical simulations.

**Essential References Not Discussed:**

See above, especially [1].

**Experimental Designs Or Analyses:**

The considered PDE problems are reasonable.

**Methods And Evaluation Criteria:**

The set of numerical examples is reasonable and the problems considered are of suitable difficulty to be interested. The comparison to existing methods in the literature is sound. The manuscript would however benefit from a quantification of the randomness (over the network initialization) in the training process. The authors might consider to do this for a subset of the experiments as an evaluation for the full set of experiments is likely unreasonably computationally expensive.

**Other Comments Or Suggestions:**

- The resolution of Figure 1 should be increased.

**Other Strengths And Weaknesses:**

Strengths:
- Convincing numerical results.

Weaknesses:
- The "conflict" with the existing literature should be discussed and ideally resolved. As mentioned above, existing sampling methods focus on hard samples, whereas the present work starts with easy samples.
- A quantification of the randomness of the training process should be provided, at least for some experiments. Is the training process more brittle than the baseline methods?

**Questions For Authors:**

See above.

**Relation To Broader Scientific Literature:**

Some related literature is not discussed: There are a number of works concerning adaptive sampling for PINNs, see for instance [1], [2] and the references therein. Interestingly, these works take a different approach, focusing on hard samples more than on easy samples via adapting the sample distribution based on the PDE residual. This seems somewhat contradictory to the presented findings and requires a thorough discussion.

Moreover, the authors of [1] also present a convincing argument for focusing on high residual/difficult regions first : For many examples — take for instance the convection equation example in [1] — the PDE residual without the boundary conditions possesses trivial minimizers, namely constant functions. Focusing on interior regions first seems therefore not reasonable.

[1] https://arxiv.org/pdf/2207.02338
[2] https://www.sciencedirect.com/science/article/abs/pii/S0045782522006260

**Theoretical Claims:**

Not applicable.

---

> ### Author Rebuttal · Authors · 2025-04-01
>
> We greatly appreciate your valuable comments. Below is our point-by-point response.
>
> **Q1: Essential References Not Discussed & Questions For Authors (1)**
>
> **R1:**
>
> 1) We will include the discussion about adaptive sampling in the Introduction and Related Work Sections of the next version. The key to adaptive sampling is to improve the learning efficiency and accuracy of the neural network by dynamically selecting or reallocating sampling points. Specifically, for [1] you mentioned, it balances regions of high and low residuals by dynamically resampling. For [2] you mentioned, it uses resampling to increase the sampling points in high PDE residual regions to avoid so-called propagation failures. We further add a review of other literature: [3] proposes a risk min–max framework to do adaptive sampling to speed up the convergence of the loss and achieve higher accuracy.
> 2)  From a technical perspective, adaptive sampling technology uses PDE residuals to adjust sample density. Our CoPINN uses the gradient of PDE residuals to estimate the learning difficulty of samples, thereby optimizing the learning process of the neural network. From a conceptual perspective, adaptive sampling technology increases the density of difficult samples while reducing the density of simple samples, thereby fully learning difficult samples. Unlike them, our CoPINN does not involve changes in sampling and sample density. CoPINN directly evaluates the learning difficulty of samples and imitates the human cognitive process to learn from easy to difficult, thereby improving accuracy. In general, adaptive sampling focuses on adjusting the number of difficult and easy samples, and CoPINN uses the early learning of simple samples to fully mine information from difficult samples in the later stage of learning. They are two different technical routes and there is no conflict.
> 3) [2]proposes that the correct solution of PINN should propagate from the boundary to the center, otherwise, it will cause trivial solutions. Therefore, PINN training should focus on the boundary. Our CoPINN doesn‘t ignore boundary conditions and initial conditions during training. Although CoPINN pays less attention to difficult samples (usually at the boundary) according to the gradient of PDE loss in the early stage of training, it does not ignore difficult samples. As learning progresses, CoPINN pays more attention to difficult samples. Therefore, CoPINN doesn't ignore difficult samples in the early stage of learning, which will not cause trivial solutions mentioned by the reviewer and is therefore reasonable.
>
> [1]A comprehensive study of non-adaptive and residual-based adaptive sampling for physics-informed neural networks.
>
> [2]Mitigating Propagation Failures in Physics-informed Neural Networks using Retain-Resample-Release (R3) Sampling.
>
> [3]A Gaussian mixture distribution-based adaptive sampling method for physics-informed neural networks.
>
> **Q2: Questions For Authors (2)**
>
> **R2:** Regarding your question, "A quantification of the randomness of the training process ", we don‘t understand what it means. Based on your comment, we guess that you are doubtful about the stability of our CoPINN. In our paper, all quantitative experiments are repeated 5 times with different random seeds, and the average values ​​are reported. To demonstrate the stability of CoPINN, below, we report the mean and standard deviation of the relative $L_2$ error on the Helmholtz dataset. The results show that CoPINN achieves both the lowest mean and standard deviation in relative $L_2$ error, and confirm that CoPINN's training process is more robust than competing methods. If we misunderstood, please clarify, and we will provide further details.
>
> |              | $16^3$            | $32^3$            | $64^3$            | $128^3$           | $256^3$           |
> | ------------ | ----------------- | ----------------- | ----------------- | ----------------- | ----------------- |
> | AHD-PINN     | $0.2108\pm0.0181$ | $0.1903\pm0.0418$ | $0.1871\pm0.0372$ | O/M               | O/M               |
> | RoPINN       | $0.4059\pm0.0353$ | $0.3338\pm0.0502$ | O/M               | O/M               | O/M               |
> | FPINN        | $0.3862\pm0.0714$ | $0.3502\pm0.0676$ | $0.3097\pm0.0580$ | O/M               | O/M               |
> | SPINN        | $0.1177\pm0.0451$ | $0.0809\pm0.0104$ | $0.0592\pm0.0316$ | $0.0449\pm0.0337$ | $0.0435\pm0.0280$ |
> | SPINN(m)     | $0.1161\pm0.0084$ | $0.0595\pm0.0113$ | $0.0360\pm0.0082$ | $0.0300\pm0.0016$ | $0.0311\pm0.0100$ |
> | CoPINN(Ours) | $0.0172\pm0.0052$ | $0.0050\pm0.0030$ | $0.0016\pm0.0006$ | $0.0007\pm0.0002$ | $0.0006\pm0.0001$ |
>
> **Q3: Other Comments Or Suggestions**
>
> **R3:** Thanks, we will increase the resolution of Figure 1 in the next version.

---

> > ### Comment · Reviewer_wy2g · 2025-04-03
> >
> > 1. Regarding quantification of the randomness of the training process: Thanks for clarifying that you ran your experiments with 5 different seed -- I have likely confused this while reading the manuscript, sorry about that. So this is not an issue then.
> >
> > 2. I appreciate that you include a discussion of the literature regarding sampling, this is helpful. I also understand that you are not resampling but re-weighting. Based on my own experience with training PINNs I know that running in trivial solutions can be a challenging issue. It would be good if you try your method on the convection equation, the example that is in [2] with $\beta=50$ is a good example to observe this. I do not care whether your method is better or worse than the baseline in [2] for this example -- I think it is important to know if it still works for such examples. I understand time might be short to do this within the discussion period, but it can certainly be done before a (possible) camera ready version.

---

> > > ### Author Response · Authors · 2025-04-05
> > >
> > > We truly appreciate the effort and attention you have given to reviewing our manuscript. Your comments were incredibly helpful, and we have taken great care in revising the manuscript to address all of your concerns.
> > >
> > > To overcome your concerns about the running trivial solutions in our CoPINN, according to your suggestions, we conduct some experiments on the convection equation with $\beta=50$ as described in the literature [1]. The experimental results are recorded as follows. From the results, the performance of our CoPINN is indeed slightly lower than the baseline Causal R3 and slightly higher than the baseline R3. However, our CoPINN significantly outperforms the other compared methods. **This demonstrates that our method remains effective in this scenario, which successfully avoids trivial solutions and provides reliable predictions.**
> > >
> > >
> > > | Method      | Relative L2 Error (%) |
> > > | ----------- | --------------------- |
> > > | Causal PINN | $72.5 \pm 3.82$       |
> > > | RAD         | $67.1 \pm 1.57$       |
> > > | R3          | $1.47 \pm 0.45$       |
> > > | Causal R3   | $1.14 \pm 0.11 $      |
> > > | Ours        | $1.31 \pm 0.63$       |
> > >
> > > Thank you again for your constructive comments, which further inspired us to study how to avoid the problem of trivial solutions in PINN. Due to the limited time, we will provide more detailed experimental details and more comprehensive experimental results (including the results of our CoPINN on the convection equation with different $\beta$) in the camera-ready version to further support the effectiveness of our method. **We hope our response could solve all of your concerns. If you have any further insights or suggestions, please feel free to share them. Due to the rebuttal mechanism of ICML, we might not be able to provide a response again. However, we will certainly incorporate any constructive feedback into the camera-ready version.**
> > >
> > > [1]Mitigating Propagation Failures in PINNs using R3 Sampling

---

### Official Review · Reviewer_TArf · 2025-03-13

**Overall Recommendation:** 4

**Summary:**

The paper proposes CoPINN, a Cognitive Physics-Informed Neural Network that addresses the Unbalanced Prediction Problem (UPP) in PINNs. UPP arises from treating easy and hard samples (e.g., boundary vs. smooth regions) equally, leading to unstable training. CoPINN introduces three key components: (1) separable subnetworks for encoding 1D coordinates to reduce computational costs, (2) dynamic difficulty evaluation of samples via PDE residual gradients, and (3) a cognitive training scheduler that progressively focuses on harder samples. Experiments on Helmholtz, Klein-Gordon, and Diffusion equations demonstrate state-of-the-art performance, with significant error reductions. The method also shows scalability to high-dimensional PDEs and robustness in boundary regions.

**Claims And Evidence:**

The claims made in the submission are supported by clear and convincing evidence.

**Essential References Not Discussed:**

No relevant works are critical to understanding the main contribution (context) of the paper, but are not currently cited/discussed in the paper.

**Experimental Designs Or Analyses:**

I checked the soundness/validity of the experimental design and analysis. In this paper, a wide range of experiments are carried out on multiple PDEs and dimensions, and the experiments and the analysis of the experiments are sound and valid.

**Methods And Evaluation Criteria:**

The proposed methods and evaluation criteria make sense for the problem at hand.

**Other Comments Or Suggestions:**

This article has some typos, including but not limited to: "Klei-Gordon" → "Klein-Gordon" (Page 14).

**Other Strengths And Weaknesses:**

Strengths:

1. This paper find the problem of existing PINN through experimental results (equal treatment of samples of different difficulty, resulting in non-optimal performance), and innovatively propose Cognitive PINN, and the experimental results prove the effectiveness of the proposed method.

2. This paper integrates self-paced learning into PINN to effectively solve the unbalanced prediction problem by dynamically prioritizing sample difficulty during training.

3. The proposed method demonstrates scalability to high-dimensional PDEs (e.g., 4D systems), overcoming memory constraints faced by baseline approaches.

4. In this paper, experimental results in different PDEs (Helmholtz, Klein-Gordon, Diffusion) show significant error reduction and robustness in the boundary region.

Weaknesses:

1. In this paper, the proposed cognitive scheduler and difficulty evaluation mechanism are not theoretically proved, so the optimality of this method has not been verified.

2. The proposed methods are sensitive to hyperparameters such as β, which presents tuning challenges, especially in balancing attention between simple and hard samples of different PDEs.

3. Progressive weighting of hard samples risks overfitting to localized boundary phenomena at the expense of global solution accuracy, especially in datasets with small difficulty variations. In the method proposed in this paper, progressive weighting of hard samples may overfit local boundary phenomena at the expense of the accuracy of the global solution, which may be more obvious in datasets with small difficulty variations.

**Questions For Authors:**

1. How does CoPINN mitigate overfitting to hard samples in later training stages? In other words, how to ensure that the prediction accuracy of easy samples does not decline in the later training period?

2. Can CoPINN scale to PDEs beyond 4D (e.g., 5D or 6D)? This is crucial for the scalability of the proposed method.

3. Why to use the IMP metric instead of standard relative improvement? This may lead to an overstatement of the experimental results of the proposed method.

**Relation To Broader Scientific Literature:**

The paper builds on PINNs (Raissi et al., 2019) and SPINN (Cho et al., 2023), integrating self-paced learning (Jiang et al., 2015) for difficulty-aware training.

**Theoretical Claims:**

I checked the correctness of proofs for theoretical claims. The difficulty measure is heuristic, and the linear weight schedule (Eq. 8–9) is intuitive.

---

> ### Author Rebuttal · Authors · 2025-04-01
>
> We sincerely thank the reviewers for their constructive feedback. Below are our responses to the questions raised:
>
> **Q1: Questions For Authors(1)**
>
> **R1:** Our proposed CoPINN employs a multi-faceted approach to prevent overfitting to hard samples during training. First, the cognitive training scheduler ensures a controlled and gradual transition of focus from easy to hard samples through the hyperparameter $\beta$, which is intentionally set to values less than 0.5 to avoid abrupt weight shifts and maintain a balanced emphasis across regions. Second, the physics-informed loss terms, specifically the initial condition loss $\mathcal L_{ic}$ and boundary condition loss $\mathcal L_{bc}$, act as implicit regularizers by enforcing physical constraints across the entire domain, thereby anchoring predictions to known conditions even as harder samples are prioritized. Finally, the robust generalization has been experimentally validated, with consistently low errors in boundary and smooth regions (e.g., Table 1, Figures 4, 9, and 10 in the original paper), confirming that the model does not overfit to specific challenging regions while maintaining accuracy across the entire solution space. This combination of gradual weighting, physics-based regularization, and experiment validation ensures stable and generalizable training.
>
> **Q2: Questions For Authors(2)**
>
> **R2:** The separable architecture of our proposed CoPINN inherently supports scalability to higher-dimensional PDEs by encoding each dimension independently through dedicated subnetworks, effectively reducing computational complexity from $O(N^d)$ to $O(dN)$, where $d$ is the number of dimensions and $N$ is the resolution per axis. While our experiments have focused on up to 4D systems (e.g., (3+1)-d Klein-Gordon) due to hardware limitations, the design principles theoretically generalize to higher dimensions. For instance, in 5D/6D scenarios, techniques like tensor decomposition (e.g., CP or Tucker formats) can optimize the aggregation of outer products in Equation 3, further enhancing efficiency. Our future work will explore these optimizations to validate scalability in extreme dimensions while maintaining the model’s accuracy and computational feasibility.
>
> **Q3: Questions For Authors(3)**
>
> **R3:** We agree that clarity in evaluation metrics is critical. The IMP metric in our paper is calculated as:
>
> $IMP=\frac{|u_s-u_b|}{u_s}\times 100\\%$
>
> where $u_s$ is the error of the suboptimal baseline and $u_b$ is the error of CoPINN. Since $u_b<u_s$ in all experiments (as CoPINN outperforms baselines), it is mathematically equivalent to the standard relative improvement formula. To eliminate ambiguity, we will revise the formula in the final version to:
> $$IMP=\frac{u_s-u_b}{u_s} \times 100 \\%$$
> removing the absolute value. This aligns with standard practice and ensures no overstatement. For example: In Table 1, a baseline error of $(0.0311)$ SPINN(m) vs. CoPINN’s $(0.0006)$ gives $IMP=\frac{0.0311-0.0006}{0.0311}\times 100\\%\approx 98\\%$, which correctly reflects the relative error reduction. In addition, we have explicitly reported both relative $L_2$ errors (e.g., Helmholtz's CoPINN: $0.0006$ vs. SPINN(m): $0.0311$), $RMSE$, and $IMP$ in Table 1 of the original paper. The performance is comprehensively demonstrated through two metrics (relative $L2$​ and $RMSE$), with the improvement magnitude illustrated via $IMP$.  We will clarify this in the text.
>
> **Q4: Other Comments Or Suggestions**
>
> **R4:** We have carefully corrected the typos in the paper:
>
> "Klei-Gordon" → "Klein-Gordon" (Page 14);
>
> "$u(x,0)=x_1+x_2, x \in \Omega, x \in \Omega$"→"$u(x,0)=x_1+x_2, x \in \Omega$" (Page 12).

---

> > ### Comment · Reviewer_TArf · 2025-04-04
> >
> > Thanks for the authors' responses. After reviewing their responses and considering the comments of other reviewers, I believe the authors have addressed my concerns effectively.
> >
> > In summary, the paper is well-organized and clearly written, with the figures aiding readers in understanding the algorithmic motivation effectively. Inspired by human cognitive learning, the proposed CoPINN is the first work to leverage self-paced learning to enhance the PINN performance in difficult regions. The technical solution is novel and reasonable. Extensive experiments are provided to make it easy to understand the contribution of the proposed method and the effectiveness of the results. Thus, I would keep my score to support my acceptance recommendation.

---

> > > ### Author Response · Authors · 2025-04-08
> > >
> > > Thanks for your support. Based on your suggestions, we will further improve the quality of our manuscript in the final version.

---

### Official Review · Reviewer_doqg · 2025-03-25

**Overall Recommendation:** 2

**Summary:**

The authors look at the PINNs setting, based on training a neural network to confirm to the PDE residual. They employ a method that dynamically samples collocation points according to the gradient of the PDE residual. They use this as a signal to do PINNs training starting with the solution on the collocation points that are easy to learn, and then move to the harder to learn regimes. They demonstrate this on four different PDEs.

**Claims And Evidence:**

The authors claim that the three categories of PINNs methods include designing the architecture of PINNs, changing the loss function, and changing the weights on the loss function for PINNs. The authors claim that their method works the best compared to various other PINNs methods on the four different PDEs.

However, they are missing a very important line of work that does adaptive sampling of the collocation points of PINNs, which is similar to what the authors are doing. For example, [1].

[1] Wu et al. A comprehensive study of non-adaptive and residual-based adaptive sampling for physics-informed neural networks. Computer Methods in Applied Mechanics and Engineering, 403, 115671, 2023.

**Essential References Not Discussed:**

This work does not discuss a broad set of PINNs literature that has focused on “adaptive sampling” techniques where different parts of the domain or different points are weighted differently, and more weight is given to points that have higher PDE residuals. Such as [1]

[1] Wu et al. A comprehensive study of non-adaptive and residual-based adaptive sampling for physics-informed neural networks. Computer Methods in Applied Mechanics and Engineering, 403, 115671, 2023.

**Experimental Designs Or Analyses:**

The authors set up 4 PDE problems to solve. They analyze this based on L2 relative error and RMSE. See above comment for questions on how reference solution data is generated and compared to.

As mentioned earlier, there isn’t a mention or comparison of the many adaptive sampling PINNs papers, such as [1]. It seems like [2] also looks at the Helmholtz problem (different source terms) through changing the loss function, and seems to mitigate this issue through loss reweighting. There are also lines of work that aims to address some of the issues the authors note, such as certain regimes being harder to learn, through imposing hard constraint [3] [4].

[1] Wu et al. A comprehensive study of non-adaptive and residual-based adaptive sampling for physics-informed neural networks. Computer Methods in Applied Mechanics and Engineering, 403, 115671, 2023.
[2] Wang et al. Understanding and mitigating gradient flow pathologies in physics-informed neural networks. SIAM Journal on Scientific Computing (2021)
[3] Lu et al. Physics-Informed Neural Networks with Hard Constraints for Inverse Design. CMAME (2021)
[4] Chalapathi et al. Scaling physics-informed hard constraints with mixture-of-experts. ICLR (2024)

**Methods And Evaluation Criteria:**

The method the authors employ is to track gradients, which is a proxy for the difficulty of learning the solution on each collocation point. They start with the model putting more weight on the easier to learn regions of the solution space, and then employ a scheduler that gradually then starts to put more weight on the harder to learn regimes. This is done in a dynamic fashion over the course of training.

As far as the evaluation, this is done by computing L2 relative error with respect to a reference solution, as well as looking at the RMSE. However, for problems that are not the diffusion problem,  it is not stated exactly what the L2 relative error is compared to: what is the reference solution that the predicted solution is compared to, and how was this data generated?

**Other Comments Or Suggestions:**

See above.

**Other Strengths And Weaknesses:**

See above. In general the bar for new PINNs methods is high, as this area has been well-studied for a number of years and there are many new methods. This method doesn’t seem to be that different from many other methods, and so any proof-of-concept should also be on harder problems.

**Questions For Authors:**

My questions are interspersed above, and some of it is summarized here:

- For problems that are not the diffusion problem,  it is not stated exactly what the L2 relative error is compared to: what is the reference solution that the predicted solution is compared to, and how was this data generated?

- This work is missing any discussion and comparison of a long line of work in doing adaptive sampling, where different parts of the domain are weighted differently (typically where PDE residuals are higher, more weight is put on this). This work is very related to what the authors are proposing. How do such  methods compare?

**Relation To Broader Scientific Literature:**

This work is part of the vast, and very well-studied, literature on PINNs and ML for solving PDEs. It proposes another method to improve the training of PINNs. Given the amount of literature in this space, there is a lot more that can be done to position this work in the broader context of the field.

**Theoretical Claims:**

There are no theoretical results.

---

> ### Author Rebuttal · Authors · 2025-04-01
>
> We sincerely thank the reviewers for their constructive feedback. Below are our responses to the questions raised:
>
> **Q1: Methods And Evaluation Criteria & Questions For Authors**
>
> **R1:** The reference solutions refer to the labels used to compute the relative $L_2$ error and $RMSE$ by comparing them with the model's predicted solutions. Following [1-3], for the Diffusion equation, reference solutions are obtained through the widely-used PDE solver platform FEniCS at a resolution of 101×101×101. For the Helmholtz and Klein-Gordon equations (3D and 4D), reference solutions with 100×100×100 resolution are independently designed and manufactured respectively.
>
> **Q2: Claims And Evidence & Experimental Designs Or Analyses & Essential References Not Discussed & Questions For Authors**
>
> **R2:** According to your suggestion, we will discuss adaptive sampling in the related work section. We acknowledge that existing work in the field of adaptive sampling, such as RAD/RAR-D [4] provides valuable insights relevant to our study. However, CoPINN fundamentally differs from these existing adaptive sampling methods:
>
> 1. Existing adaptive sampling methods like RAD/RAR-D [4] primarily focus on the global residual distribution, improving overall accuracy by resampling or adding points to the region near the high-residual points, but they ignore the Unbalanced Prediction Problem (UPP) of PINN. In contrast, the core objective of CoPINN is to resolve the UPP commonly observed in traditional PINNs near physical boundaries. We find that relying solely on absolute residual values (e.g., as in RAD) may overlook the high-gradient nature of boundary regions, leading to local overfitting or underfitting. Therefore, CoPINN introduces a residual gradient-based dynamic difficulty assessment to more accurately identify abrupt changes in boundary regions (e.g., shocks and singularities).
>
> 2. Existing adaptive methods (e.g., RAR-D) employ a greedy strategy to incrementally add high-residual points but do not consider the model’s optimization capability at different training stages. To address this, CoPINN incorporates a cognitive training scheduler, which prioritizes simpler regions (low-gradient areas) in the early stages to stabilize training, while progressively increasing the weight of more complex regions (high-gradient areas) in later stages. This prevents the model from prematurely converging to suboptimal local solutions.
>
> **Q3: Experimental Designs Or Analyses**
>
> **R3:** The reviewer notes that prior work [5] addresses similar issues via loss reweighting. However, our CoPINN fundamentally differs in both methodology and scope. [5] focuses on loss-term balancing (e.g., PDE residual vs. boundary loss) without considering spatial/temporal variations in sample difficulty. In contrast, inspired by the human curriculum learning strategy of progressing from easy to hard tasks, CoPINN introduces a cognitive learning approach. It evaluates sample-level difficulty (via PDE residual gradients) and progressively prioritizes harder regions during training. CoPINN is finer-grained, addressing intra-term imbalances (e.g., stubborn points vs. smooth regions).
>
> **Q4: Experimental Designs Or Analyses**
>
> **R4:** There are two key differences between the approach with hard constraints [6-7] and our proposed CoPINN.
>
> 1. [6-7] focus on inverse problems (e.g., geometric parameter optimization), with the core objective of simultaneously satisfying PDE constraints and optimization objectives. In contrast, CoPINN targets the Unbalanced Prediction Problem (UPP) in forward PDE solving, aiming to address error caused by varying sample difficulties in PINN training. Their application scenarios are fundamentally distinct.
>
> 2. [6-7] employ fixed hard constraints (e.g., modified network architectures) and optimization strategies to ensure strict adherence to physical properties during the prediction process, enhancing accuracy through the incorporation of additional constraints. CoPINN introduces a dynamic cognitive scheduler that evaluates sample difficulty via PDE residual gradients and progressively adjusts training weights from easy to hard samples, constituting a data-driven adaptive learning paradigm. This training strategy, inspired by human cognitive processes, remains unexplored in the PINN domain.
>
> [1] Physics-informed neural networks: A deep learning framework for solving forward and inverse problems involving nonlinear partial differential equations.
>
> [2] Gradient-enhanced physics-informed neural networks for forward and inverse PDE problems.
>
> [3] Separable physics-informed neural networks.
>
> [4]A comprehensive study of non-adaptive and residual-based adaptive sampling for physics-informed neural networks
>
> [5]Understanding and mitigating gradient flow pathologies in physics-informed neural networks.
>
> [6]Physics-informed neural networks with hard constraints for inverse design.
>
> [7]Scaling physics-informed hard constraints with mixture-of-experts.

---

> > ### Comment · Reviewer_doqg · 2025-04-02
> >
> > Thank you for the response. This response doesn't address some of my key concerns including comparisons to some of these important and related areas, as well as studies on harder problems (many of the problems studied in this paper have been previously well-studied by PINNs approaches, as was mentioned in my original comment). Showing both clear comparisons of this approach vs. the others, as well as analysis on if and why this approach indeed works better (how does the learning change, where error ends up concentrating) would make this paper stronger.
> >
> > As it stands now, I don't think this paper stands out that much from the vast literature on PINNs and methods to make PINNs converge better.

---

> > > ### Author Response · Authors · 2025-04-04
> > >
> > > We appreciate your helpful comments and will add the comparison and analysis to the next version of our paper, and make comprehensive revisions based on the above important discussions. Thanks again for your valuable suggestions and comments.
> > >
> > > **Q1: Comparisons to some of these important and related areas, as well as studies on harder problems.**
> > >
> > > We first emphasize that the proposed CoPINN is fundamentally different from these related approaches [1-4] in both the technical framework and research problem. **(1) In terms of the framework,** we propose a novel cognitive PINN method, which first prioritizes more on easy samples and gradually focuses on more challenging samples, thereby enhancing the model’s generalization to difficult samples. To the best of our knowledge, our CoPINN could be the first work that leverages self-paced learning to enhance the PINN performance in difficult regions. **In addition, the novelty of our approach is also recognized by the other three reviewers.** **(2) In terms of the research problem,** although some adaptive sampling methods address the so-called harder problem, they cannot handle the unbalanced prediction problem under the limited data conditions. Moreover, when facing high-dimensional PDEs, they suffer from high computational costs. **Therefore, the problem studied in this paper has not been well-studied by these previous PINN approaches.**
> > >
> > > Then, **to better clarify the differences between CoPINN and reference methods,** we provide the following detailed description, which will be updated in our final version.
> > >
> > > (1) Typical adaptive methods such as RAD and RAR-D [1] require dynamic sampling or addition of new data points during training, particularly in high-residual regions. These methods become inapplicable when data cannot be updated. For instance, when only fixed predefined points are available, their core premise relies on optimizing training through sampling distribution adjustments. Moreover, adaptive sampling techniques face that the number of required sampling points grows exponentially with dimensionality, rendering dynamic distribution adjustments computationally prohibitive. This limitation explains why such methods have only been tested on up to 2D datasets. In contrast, CoPINN optimizes the model by dynamically adjusting sample weights without altering data point locations or quantities. This enables performance improvements even with non-updatable fixed data points. Furthermore, our CoPINN proposes Separable Learning to reduce cross-dimensional computational coupling and avoid high-dimensional tensors, thereby significantly decreasing parameter counts and complexity. Thus, our CoPINN could be more effectively extended to high-dimensional spaces.
> > >
> > > (2) Loss reweighting methods [2] perform coarse adjustments at the loss-term level, failing to address prediction imbalances caused by per-sample difficulty variations. While they balance gradients across different loss terms through global weights, they neglect local heterogeneity among samples within the same loss term. Compared to this, CoPINN achieves fine-grained optimization through sample-level difficulty assessment and dynamic weight scheduling, thereby mining more information from complex regions such as abrupt transition zones.
> > >
> > > (3) In physical systems, certain regions like shock waves and singularities exhibit drastic variations in physical quantities, resulting in large PDE residual gradients and heightened learning challenges, while smoother regions remain relatively simple. Hard-constrained methods [3-4] uniformly process all samples without dynamic prioritization adjustments, leading to suboptimal performance in challenging areas. By contrast, CoPINN quantifies sample difficulty using PDE residual gradients to identify stubborn regions. Through cognitive training scheduling, it first assigns higher weights to simpler samples to stabilize training, then gradually increases weights for harder samples to optimize performance in challenging zones.
> > >
> > > **Q2: How does the learning change?**
> > >
> > > The learning process of our CoPINN is as follows. During each epoch, CoPINN first dynamically evaluates each sample's difficulty based on gradients of PDE residuals. Then, we adaptively assign greater weights to easy samples in the early stage of training, and assign greater weights to difficult samples in the later stage of training (see Fig.3(c)). This learning scheme gradually shifts the neural network’s focus from simpler samples to more challenging ones.
> > >
> > > **Q3: Where error ends up concentrating?**
> > >
> > > To better clarify where the errors of CoPINN ultimately concentrate, you can refer to Figures 1, 4, 9, and 10 in the original paper. These figures demonstrate that, compared to baselines, CoPINN successfully reduces prediction errors in stubborn regions (areas with abrupt changes) and eliminates observable error concentration zones. Therefore, at the end of training, the errors of our CoPINN are not concentrated.

---

### Decision · Program_Chairs · 2025-05-01

**Decision:**

Accept (spotlight poster)

**Comment:**

CoPINN presents a compelling and original contribution to the PINNs landscape. The paper is well-motivated, methodologically sound, and thoroughly evaluated. It introduces self-paced learning in the context of PDE solving, a conceptually novel direction that offers substantial improvements over existing methods, particularly in handling difficult regions like boundaries or singularities.

The rebuttal was professional, detailed, and addressed every reviewer concern effectively, including additional experiments where necessary. The work is highly relevant not only to PINNs, but also to broader efforts in difficulty-aware training and adaptive learning strategies in scientific machine learning.